# Jailbreak to Protect: Buffering and Reinforcing via Temporary Jailbreaking for Safe Fine-Tuning in Large Language Models

**Seokil Ham**[1]  **Jaehyuk Jang**[1]  **Wonjun Lee**[1]  **Changick Kim**[1]

## Abstract

Fine-tuning-as-a-Service (FaaS) enables personalization of large language models (LLMs), but it can weaken safety-alignment under harmful fine-tuning attacks. Recent work has shown that activating harmful-behavior modules during fine-tuning can prevent models from learning undesired behaviors, but its mechanism remains unclear. In this paper, we revisit temporary jailbreaking as a defense against harmful fine-tuning and provide a gradient-level analysis showing that it saturates safety-degrading gradients while preserving benign task-relevant gradients. Based on this insight, we propose a **Buffer-and-Reinforce fine-tuning framework** that buffers harmful updates during user fine-tuning and reinforces safety after adaptation. Specifically, BufferLoRA induces temporary jailbreaking as a removable adapter to reduce harmful updates during user fine-tuning. After adaptation, ReinforceLoRA, trained to recover refusal behavior under the temporarily jailbroken state, is integrated with User-LoRA via QR decomposition-based merging to reinforce safety while preserving user-task performance. Extensive experiments show that our framework achieves superior safety and utility with no additional safety data during user fine-tuning and minimal computational cost.

## 1. Introduction

Recently, AI service providers such as OpenAI and Google have introduced Fine-tuning-as-a-Service (FaaS), which allows users to customize pre-trained large language models (LLMs) by fine-tuning them on user-provided datasets. This service addresses the growing demand for domain-

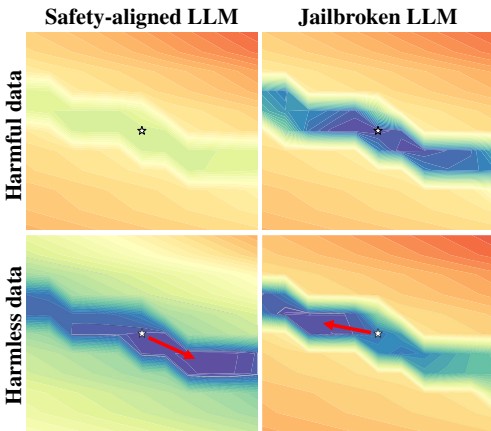

*Figure 1.* **2D Loss landscapes of a safety-aligned and a jailbroken LLM, evaluated on harmful and harmless data.** Warmer (cooler) regions indicate higher (lower) loss, and the star marks (☆) the current model parameters. The jailbroken LLM has largely converged on harmful data, whereas the safety-aligned LLM has not, while both models maintain room for optimization on harmless data, as indicated by the red arrow. The loss landscapes are obtained using a 2D random-plane projection around the safety-aligned and jailbroken LLaMA3-8B-Instruct parameter points.

specific or personalized LLMs tailored to users' objectives and data distributions. However, FaaS also introduces significant safety risks, as personalized LLMs may be misused for safety-critical purposes such as criminal planning or weapon construction. Recent studies (Qi et al., 2024; Lermen et al., 2024) show that fine-tuning often leads to substantial safety degradation, even when starting from safety-aligned LLMs. This degradation is further exacerbated when user data contains harmful data. Attacks that exploit this vulnerability by injecting harmful prompts into the fine-tuning dataset are referred to as *harmful fine-tuning attacks*.

Prior work on defending against these attacks is categorized into three classes: alignment-stage (Rosati et al., 2024; Huang et al., 2024b; 2025b; Liu et al., 2025a), fine-tuning-stage (Mukhoti et al., 2024; Bianchi et al., 2024; Huang et al., 2024a; Li et al., 2025a; Yang et al., 2026; Ham et al., 2025), and post-fine-tuning-stage approaches (Hsu et al., 2024; Huang et al., 2025a; Wang et al., 2026). Many of these methods prevent harmful updates by explicitly constraining

---

[1]School of Electrical Engineering, Korea Advanced Institute of Science and Technology (KAIST), Daejeon, Republic of Korea. Correspondence to: Changick Kim <changick@kaist.ac.kr>.

*Proceedings of the 43rd International Conference on Machine Learning*, Seoul, South Korea. PMLR 306, 2026. Copyright 2026 by the author(s).

the optimization process through regularization-based mechanisms. However, this paradigm introduces several practical limitations during user fine-tuning, including reliance on additional safety-alignment data, increased computational overhead, and limited effectiveness in mitigating harmful updates.

Beyond these regularization-based defenses, a recent work (Zhou et al., 2024) has explored a different strategy: activating harmful-behavior modules during fine-tuning to prevent models from further learning undesired behaviors. However, the mechanism underlying this effect remains insufficiently understood. In this paper, we revisit temporary jailbreaking as a defense against harmful fine-tuning and provide a gradient-level analysis. Our analysis shows that, once an LLM is temporarily jailbroken, safety-degrading gradients become largely saturated, while benign task-relevant gradients remain active during fine-tuning. Figure 1 illustrates the loss landscapes of safety-aligned LLM (LLaMA3-8B-Instruct (Grattafiori et al., 2024)) and jailbroken LLM (LLaMA3-8B-Instruct fine-tuned on the LAT harmful data (Sheshadri et al., 2024) via LoRA (Hu et al., 2022)) evaluated on harmful (Ji et al., 2023) and harmless data (Cobbe et al., 2021). When the star marks denote the current model parameters, the jailbroken model is already converging on harmful data, whereas a safety-aligned model exhibits higher loss and thus significant capacity for further jailbreaking. In contrast, for harmless data, both models retain room for further optimization, as indicated by the red arrow. This analysis suggests that harmful updates can be mitigated not by explicit regularization, but by exploiting the model's convergence during fine-tuning.

Building on this insight, we propose a **Buffer-and-Reinforce fine-tuning framework** that buffers harmful updates by temporarily jailbreaking the model during user fine-tuning and reinforces safety after user fine-tuning. Specifically, we prepare two auxiliary LoRA modules (Hu et al., 2022) before user fine-tuning: **BufferLoRA**, which induces temporary jailbreaking, and **ReinforceLoRA**, which is trained to recover safe refusal behavior even under the temporarily jailbroken state. During user fine-tuning, Buffer-LoRA is attached to the base LLM and kept frozen to induce jailbreaking, while a **UserLoRA** is simultaneously attached and trained on user data. This design prevents harmful updates from being encoded into the UserLoRA while allowing UserLoRA to capture benign user-task-relevant knowledge. After fine-tuning, BufferLoRA is removed, and ReinforceLoRA is integrated with UserLoRA via QR decomposition-based orthogonal merging, which suppresses components that interfere with the user-task subspace. This post-training reinforcement strengthens the safety of the personalized model while preserving user-task performance.

Extensive experiments demonstrate that our framework achieves both strong safety and user-task performance with no additional safety-alignment data and minimal computational cost, making it a practical solution for secure FaaS.

## 2. Related Works

**Safety in Large Language Models.** As LLMs are increasingly deployed in real-world applications, ensuring their safety against malicious use and unintended harmful behaviors has become a central concern. Existing approaches to LLM safety can be broadly grouped into three categories. The first category focuses on input-output level filtering, where harmful queries or responses are detected and blocked using external classifiers (Inan et al., 2023; Rebedea et al., 2023) or internal signals that distinguish harmful from benign content (Zheng et al., 2024; Han et al., 2025; Ham et al., 2025; Zhao et al., 2026). The second line of work adopts adversarial-training-based defenses, which explicitly expose models to jailbreaking prompts and train them to refuse harmful requests (Sheshadri et al., 2024; Xhonneux et al., 2024; Zou et al., 2024; Yu et al., 2025). More recently, a third paradigm has emerged that leverages the reasoning capabilities of LLMs to make safety-aware decisions (Zhang et al., 2025; Kang & Li, 2025; Jiang et al., 2025; Jeung et al., 2026). In contrast, we focus on harmful fine-tuning attacks and propose a buffer-and-reinforce framework that mitigates safety degradation during user fine-tuning and strengthens the safety of personalized LLMs after fine-tuning.

**Defending Harmful Fine-tuning Attacks.** Harmful fine-tuning attacks jailbreak a safety-aligned LLM by injecting harmful query-response pairs into the fine-tuning dataset, causing safety degradation even with small amounts of harmful or purely benign data (Qi et al., 2024; Lermen et al., 2024; Hsiung et al., 2025). These findings highlight the importance of robust and practical defenses for safe fine-tuning. Existing defenses fall into three classes: alignment-stage methods focus on creating initial model weights resistant to safety degradation (Rosati et al., 2024; Huang et al., 2024b; Liu et al., 2025a; Huang et al., 2025b; Liu et al., 2025b; Perin et al., 2025; Chen et al., 2025), fine-tuning-stage methods prevent harmful updates by imposing constraints during fine-tuning (Mukhoti et al., 2024; Bianchi et al., 2024; Huang et al., 2024a; Li et al., 2025b;a; Ham et al., 2025; Yang et al., 2026), and post-fine-tuning-stage methods detect and remove harmful components after fine-tuning (Hsu et al., 2024; Du et al., 2024; Huang et al., 2025a; Lu et al., 2025; Yi et al., 2025; Wang et al., 2026). Closest to our work, Security Vector (Zhou et al., 2024) activates harmful-behavior modules during fine-tuning to make undesired behaviors unlearnable, but does not provide a mechanistic analysis of why this strategy works. In contrast, we analyze temporary jailbreaking at the gradient level and show that it saturates safety-degrading gradients while preserving be-

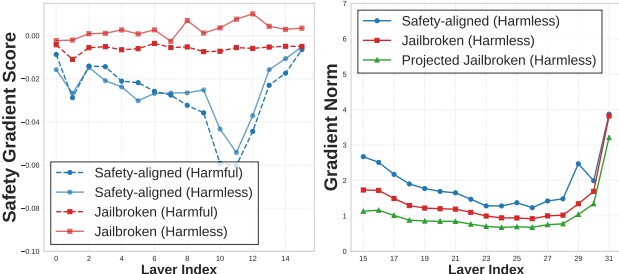

*Figure 2.* **Safety Gradient Score and Gradient Norms of safety-aligned and jailbroken LLMs on harmful and harmless data.** *Projected Jailbroken* denotes the gradient of the jailbroken LLM projected onto the gradient direction of the safety-aligned LLM. Jailbroken LLMs exhibit reduced susceptibility to harmful updates while maintaining comparable learning capacity on harmless data.

nign task-relevant gradients. Building on this insight, we use temporary jailbreaking not only to buffer harmful updates during user fine-tuning but also to train ReinforceLoRA for post-fine-tuning safety reinforcement.

## 3. Problem Setting: Scenario & Threat Models

In FaaS, AI providers pursue two primary objectives: (i) achieving high user-specific task performance and (ii) preserving safety-alignment. To formalize this setting, we specify the data available to each party. The FaaS provider has access to a curated safety dataset consisting of 5,000 harmful queries paired with both harmful and refusal responses, and 5,000 harmless queries paired with helpful responses. In contrast, users submit their own datasets for model personalization via fine-tuning. We model the user data as a mixture containing $p \in [0, 1]$ harmful queries paired with harmful responses and $(1 - p)$ harmless queries paired with helpful responses sampled from the same benign data distribution. We explicitly assume that the harmful prompt distribution in the user data differs from that of the provider's safety dataset. Crucially, the provider has no prior knowledge of whether the user data contains harmful queries or of its underlying data distribution, making defenses against harmful fine-tuning attacks particularly challenging.

## 4. How Temporary Jailbreaking Buffers Harmful Updates

Figure 1 shows that jailbreaking an LLM leads to loss convergence on harmful data, suggesting saturation of harmful gradients. However, it remains unclear whether this convergence actually corresponds to a reduction in gradients that directly degrade the LLM safety. To investigate this, we introduce a novel gradient-based metric, the *Safety Gradient Score*, which quantifies the extent of safety degradation induced by harmful and harmless data for both safety-aligned and jailbroken LLMs. Based on prior work suggest-

ing the existence of safety-related directions in the parameter space (Hsu et al., 2024; Li et al., 2025a; Yang et al., 2026), we evaluate safety degradation by measuring the magnitude of gradients that conflict with the safety directions. We compute the Safety Gradient Score on layers 0 to 15, where the safety-gradient contrast is most pronounced, following prior work that associates safety-related behavior with early-to-middle layers (Li et al., 2025b; Arditi et al., 2024; Ham et al., 2025). This boundary is specific to LLaMA3-8B-Instruct and is not intended as a universal boundary across LLMs. Scores are averaged over 1,000 samples, using BeaverTails (Ji et al., 2023) and GSM8K (Cobbe et al., 2021) as the harmful and harmless datasets, respectively. Formally, we define the Safety Gradient Score $S$ as the dot product between the gradient $\mathbf{g}$ and the safety direction $\mathbf{v}$. For each layer $l$, the score is computed as follows:

$$S^l = \frac{1}{N} \sum_{i=1}^{N} \frac{\mathbf{g}_i^l \cdot \mathbf{v}^l}{\|\mathbf{v}^l\|_2 + \epsilon}, \tag{1}$$

where $N$ is the number of data samples and $\epsilon = 10^{-8}$ prevents division by zero. The safety direction $\mathbf{v}^l$ is derived from the LoRA weights obtained by safety-aligning the LLM. Intuitively, a negative score indicates a direction that degrades safety, while a positive score implies a direction that reinforces safety. Figure 2 shows that, for the safety-aligned LLM, fine-tuning on either harmful or harmless data yields negative scores across these layers, indicating that standard fine-tuning compromises safety regardless of data harmfulness. In contrast, the jailbroken LLM exhibits Safety Gradient Scores near zero for both data types, suggesting that the safety-degrading gradients are largely saturated.

Furthermore, to evaluate whether the jailbroken LLM retains the ability to learn from harmless data, we analyze gradient norms on harmless samples. Following prior work suggesting that model utility is primarily determined by later layers (Chuang et al., 2024), we focus on layers 15 and above, where utility-related gradients are more pronounced in LLaMA3-8B-Instruct. This model-specific partition is used only for analysis and does not affect the fine-tuning procedure. Gradients are averaged over 1,000 GSM8K samples. Figure 2 shows that the jailbroken LLM exhibits gradient norms comparable to those of the safety-aligned LLM, highlighting its preserved capacity to learn user tasks. Beyond gradient norms, we further examine whether the utility-relevant gradient directions are preserved. Since no oracle exists for the optimal utility gradient, we use the safety-aligned model's gradient as a proxy and define *Projected Jailbroken* as the dot product between the jailbroken LLM's gradient and the normalized safety-aligned LLM's gradient. The projected magnitude closely matches the original gradient magnitude, indicating strong directional alignment.

In conclusion, our analysis shows that fine-tuning a jailbroken LLM effectively mitigates safety-degrading gradients,

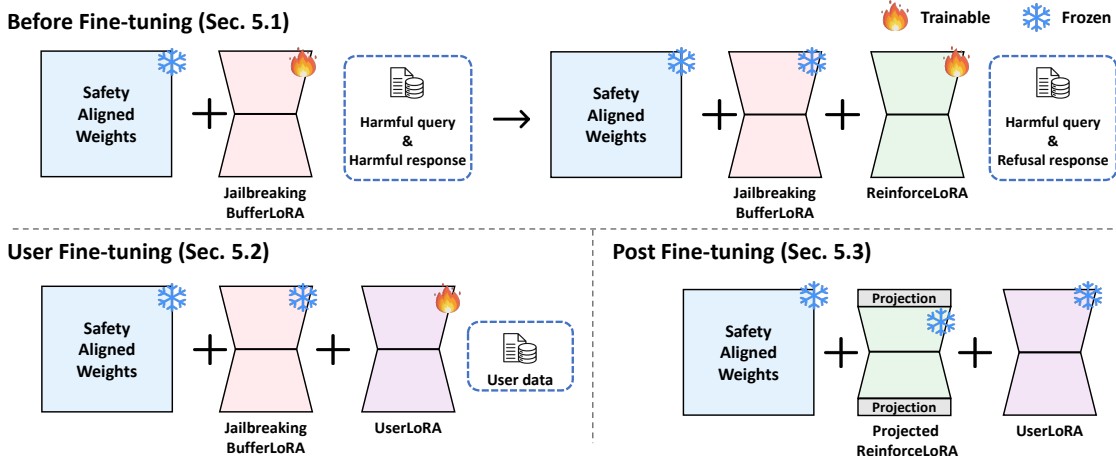

*Figure 3.* **Overview of the Buffer-and-Reinforce fine-tuning framework.** Before fine-tuning, BufferLoRA and ReinforceLoRA are prepared in advance by the FaaS provider. During fine-tuning, the model is temporarily jailbroken via BufferLoRA, and only the UserLoRA is trained on user data, with harmful updates mitigated by BufferLoRA. After fine-tuning, ReinforceLoRA is orthogonally projected with respect to the UserLoRA, and the projected ReinforceLoRA is merged with the UserLoRA and added to the original safety-aligned LLM.

regardless of data harmfulness, while preserving the ability to learn from harmless data. Moreover, when temporary jailbreaking is implemented via an attachable LoRA module, the LLM's original safety-alignment can be restored by removing the LoRA after fine-tuning (Zhou et al., 2024). These observations motivate our **Buffer-and-Reinforce fine-tuning framework**, where **BufferLoRA** buffers harmful updates during user fine-tuning and **ReinforceLoRA** further strengthens the fine-tuned model's safety.

## 5. Methodology

We propose a **Buffer-and-Reinforce framework** for safe fine-tuning using three modular components: (i) **BufferLoRA** for temporary jailbreaking, (ii) **ReinforceLoRA** for safety reinforcement, and (iii) **UserLoRA** for user-specific adaptation. Before user fine-tuning, BufferLoRA and ReinforceLoRA are prepared in advance. During user fine-tuning, BufferLoRA is attached to the LLM and kept frozen, while only UserLoRA is trained on user data. After fine-tuning, BufferLoRA is removed, and ReinforceLoRA is merged with UserLoRA via QR decomposition-based orthogonal merging to obtain the final personalized model. This design requires no additional safety-alignment data during user fine-tuning and incurs minimal computational overhead. An overview of the framework is shown in Figure 3.

### 5.1. Before Fine-tuning

**BufferLoRA** BufferLoRA is a LoRA module that induces a temporarily jailbroken state to drive loss convergence on harmful data. While temporary harmful-behavior activation has been explored in prior work (Zhou et al., 2024), our use of BufferLoRA is grounded in the gradient-level analysis in

Section 4, which shows that temporary jailbreaking saturates safety-degrading gradients while preserving benign task-relevant gradients. Unlike prior work, which relies on a KL loss on harmless data to preserve benign task performance, our analysis allows us to remove the KL loss. The training objective for BufferLoRA parameters $\theta_B$ is defined as:

$$\mathcal{L}_B(\theta_B) = -\mathbb{E}_{(x,y)\sim\mathcal{D}_H}\left[\sum_{t=1}^{|y|}\log P(y_t \mid x, y_{<t}; \theta, \theta_B)\right], \quad (2)$$

where $\mathcal{D}_H$ contains harmful queries $x$ paired with harmful responses $y$, $\theta$ denotes the frozen base LLM parameters, and $P(\cdot)$ denotes the token probability. Optimizing this objective encourages the BufferLoRA-attached LLM to generate harmful responses for harmful queries. A key advantage of BufferLoRA is its attachable and removable nature. We attach BufferLoRA only during user fine-tuning to buffer against harmful updates, then remove BufferLoRA afterward to restore the base model's original safety-aligned behavior. Importantly, BufferLoRA is trained only once before service deployment and reused across all user fine-tuning sessions. As a result, this avoids repeated jailbreaking overhead during user fine-tuning while providing a consistent mechanism for mitigating harmful fine-tuning attacks.

**ReinforceLoRA** While BufferLoRA prevents safety degradation, it only maintains the base LLM's safety, which may be imperfect. To further reinforce safety, we incorporate ReinforceLoRA, a LoRA module designed to strengthen the safety of the final personalized model after user fine-tuning. Similar to post-fine-tuning-stage methods such as Panacea (Wang et al., 2026), ReinforceLoRA is optimized before being applied after user fine-tuning. Unlike Panacea, which learns an adaptive perturbation to increase harmful

loss, ReinforceLoRA is trained under the temporarily jail-broken state induced by BufferLoRA to recover refusal behavior and reinforce safety after adaptation. Specifically, the LLM is first jailbroken using BufferLoRA, and ReinforceLoRA is optimized to produce refusal responses to harmful queries while keeping both the base model and BufferLoRA frozen. The training loss for ReinforceLoRA parameters $\theta_R$ is defined as:

$$\mathcal{L}_R(\theta_R) = -\mathbb{E}_{(x,y)\sim\mathcal{D}_S\cup\mathcal{D}_B}\left[\sum_{t=1}^{|y|}\log P(y_t \mid x, y_{<t};\theta,\theta_B,\theta_R)\right],$$
(3)

where $\mathcal{D}_S$ contains harmful queries paired with refusal responses, and $\mathcal{D}_B$ contains benign queries paired with appropriate responses. Training ReinforceLoRA only on $\mathcal{D}_S$ causes collapse to refusing all inputs, so we jointly train on $\mathcal{D}_S$ and $\mathcal{D}_B$ to reject only harmful queries while preserving benign behavior. Like BufferLoRA, ReinforceLoRA is trained once before deployment and reused across all user fine-tuning sessions, incurring minimal overhead.

## 5.2. User Fine-tuning

During user fine-tuning, we exploit the harmful loss convergence induced by BufferLoRA to safely learn user-specific knowledge. Specifically, we attach BufferLoRA to the base LLM for jailbreaking, and train only a UserLoRA on user data while keeping the base model and BufferLoRA frozen. The objective for UserLoRA parameters $\theta_U$ is defined as:

$$\mathcal{L}_U(\theta_U) = -\mathbb{E}_{(x,y)\sim\mathcal{D}_U}\left[\sum_{t=1}^{|y|}\log P(y_t \mid x, y_{<t};\theta,\theta_B,\theta_U)\right],$$
(4)

where $\mathcal{D}_U$ denotes the user data, which may contain an unknown mixture of harmless and harmful data. UserLoRA training with BufferLoRA captures primarily benign, task-relevant knowledge from the user data, even when harmful data exists. During this process, BufferLoRA acts as a buffer that saturates safety-degrading gradients through harmful loss convergence. After user fine-tuning, BufferLoRA is removed to restore the base LLM's original safety-alignment, yielding a model that incorporates user-specific knowledge without introducing additional jailbreaking behavior.

## 5.3. Post-fine-tuning

After user fine-tuning and removal of BufferLoRA, the final model is constructed by combining UserLoRA for downstream tasks with ReinforceLoRA for safety reinforcement. However, naively merging ReinforceLoRA with UserLoRA can cause destructive interference, distorting user-specific representations and degrading user-task performance.

To mitigate this issue, we adopt a **QR decomposition-based merging strategy** that integrates only the components of ReinforceLoRA that are orthogonal to the user-task subspace. Let $W_U = B_U A_U$ and $W_R$ denote the weights of UserLoRA and ReinforceLoRA, respectively, where $B_U \in \mathbb{R}^{d_{\text{out}}\times r}$ and $A_U \in \mathbb{R}^{r\times d_{\text{in}}}$ are the output and input LoRA matrices of UserLoRA. Rather than performing decomposition on the full matrix $W_U$, we exploit the low-rank structure of LoRA by identifying the user-task subspace via QR decomposition of $B_U$. Appendix B.1 formally proves that span$(W_U)$ is well approximated by span$(B_U)$, justifying this approximation. Specifically, we compute an orthonormal basis $Q_B$ and project ReinforceLoRA onto its orthogonal complement.

However, a potential challenge is *rank collapse*, where the effective rank of UserLoRA is lower than its nominal rank $r$, which causes over-removal of ReinforceLoRA components. To address this, we optionally restrict the projection to the effective UserLoRA subspace, identified via eigen-decomposition of the Gram matrix $G = A_U A_U^\top \in \mathbb{R}^{r\times r}$:

$$G v_i = \lambda_i v_i, \quad \text{s.t.} \quad \lambda_i > \tau \cdot \max_j \lambda_j,$$
(5)

$$V_{\text{eff}} = [v_1, \ldots, v_k] \in \mathbb{R}^{r\times k},$$
(6)

where $\lambda_i$ and $v_i$ denote the $i$-th eigenvalue and eigenvector of $G$, respectively. $\tau$ is a threshold used to determine the effective rank $k$. Appendix B.2 formally proves that applying this restriction yields an output subspace that is identical to the subspace obtained from the rank-$k$ truncation of $A_U$.

Using this effective subspace, we form $\hat{B}_U = B_U V_{\text{eff}}$, obtain an orthonormal basis $Q_B$ via QR decomposition, and use $Q_B$ to compute the final model weights as:

$$\hat{B}_U = Q_B R,$$
(7)

$$\tilde{W}_R = (I - \alpha Q_B Q_B^\top)W_R,$$
(8)

$$W_{\text{final}} = W_{\text{base}} + \frac{1}{2}\left(W_U + \tilde{W}_R\right),$$
(9)

where $\alpha$ controls the strength of orthogonalization. Since $W_U$ and $\tilde{W}_R$ are both residual LoRA updates, we average them to prevent the merged update from having an inflated magnitude while preserving the balance between user adaptation and safety reinforcement. In practice, UserLoRA typically retains full effective rank, implying span$(W_U) \approx$ span$(B_U)$. Thus, we apply the effective-rank restriction only when rank collapse is detected. To further mitigate over-removal, we use soft orthogonalization via $\alpha$ rather than hard projection, selectively suppressing task-interfering updates while preserving safety and utility.

As a result, the final model weights encode (i) user-specific knowledge from UserLoRA and (ii) safety reinforcement from ReinforceLoRA, without re-introducing jailbreaking behavior or compromising downstream task performance.

*Table 1.* **Comparison of safety and utility across varying harmful data ratios ($p$), ranging from 0.0 to 1.0.** All results are averaged over three seeds (30, 42, and 50). Fine-tuning Accuracy is omitted for $p = 1.0$ due to the absence of harmless data. Our fine-tuning framework exhibits superior robustness, achieving low Harmful Score and high Fine-tuning Accuracy across all ratios.

| Methods | Harmful Score (↓) | | | | | Fine-tuning Accuracy (↑) | | | | |
| --- | --- | --- | --- | --- | --- | --- | --- | --- | --- | --- |
| | $p = 0.0$ | $p = 0.1$ | $p = 0.3$ | $p = 0.5$ | $p = 1.0$ | $p = 0.0$ | $p = 0.1$ | $p = 0.3$ | $p = 0.5$ | $p = 1.0$ |
| SFT | $33.0_{\pm 1.0}$ | $75.2_{\pm 0.5}$ | $79.1_{\pm 0.1}$ | $80.7_{\pm 0.5}$ | $81.0_{\pm 0.2}$ | $70.6_{\pm 0.8}$ | $69.0_{\pm 0.9}$ | $67.4_{\pm 0.4}$ | $67.3_{\pm 1.7}$ | - |
| LDIFS (Mukhoti et al., 2024) | $16.6_{\pm 0.5}$ | $16.4_{\pm 0.9}$ | $16.5_{\pm 0.4}$ | $16.9_{\pm 1.2}$ | $17.6_{\pm 0.3}$ | $75.4_{\pm 0.3}$ | $74.1_{\pm 1.1}$ | $73.0_{\pm 1.3}$ | $73.5_{\pm 0.6}$ | - |
| SafeInstruct (Bianchi et al., 2024) | $\mathbf{7.9}_{\pm 1.1}$ | $19.6_{\pm 1.6}$ | $50.5_{\pm 3.5}$ | $66.3_{\pm 1.8}$ | $74.0_{\pm 0.8}$ | $70.4_{\pm 1.3}$ | $69.4_{\pm 0.5}$ | $67.8_{\pm 0.3}$ | $67.2_{\pm 0.2}$ | - |
| Lisa (Huang et al., 2024a) | $14.7_{\pm 0.5}$ | $29.9_{\pm 2.7}$ | $49.9_{\pm 7.7}$ | $64.0_{\pm 6.0}$ | $73.5_{\pm 2.4}$ | $70.0_{\pm 0.7}$ | $69.5_{\pm 0.4}$ | $67.6_{\pm 1.8}$ | $66.9_{\pm 3.4}$ | - |
| Security Vector (Zhou et al., 2024) | $24.3_{\pm 0.3}$ | $22.1_{\pm 0.7}$ | $22.8_{\pm 0.7}$ | $25.3_{\pm 0.9}$ | $23.9_{\pm 1.3}$ | $72.3_{\pm 0.8}$ | $71.3_{\pm 0.8}$ | $69.3_{\pm 0.4}$ | $68.7_{\pm 0.9}$ | - |
| AsFT (Yang et al., 2026) | $21.1_{\pm 0.6}$ | $26.7_{\pm 0.7}$ | $29.6_{\pm 0.9}$ | $32.2_{\pm 0.8}$ | $34.1_{\pm 0.2}$ | $67.0_{\pm 0.2}$ | $68.4_{\pm 0.1}$ | $68.9_{\pm 1.7}$ | $69.3_{\pm 0.8}$ | - |
| SafeLoRA (Hsu et al., 2024) | $20.0_{\pm 0.2}$ | $26.6_{\pm 0.8}$ | $31.7_{\pm 0.8}$ | $41.6_{\pm 1.5}$ | $53.2_{\pm 0.8}$ | $75.1_{\pm 0.7}$ | $73.3_{\pm 0.5}$ | $73.4_{\pm 0.4}$ | $73.1_{\pm 0.4}$ | - |
| Antidote (Huang et al., 2025a) | $22.2_{\pm 3.4}$ | $27.2_{\pm 1.7}$ | $45.4_{\pm 19.8}$ | $49.8_{\pm 20.1}$ | $58.1_{\pm 14.2}$ | $\mathbf{76.1}_{\pm 0.7}$ | $75.0_{\pm 0.3}$ | $74.2_{\pm 1.6}$ | $74.7_{\pm 2.2}$ | - |
| Panacea (Wang et al., 2026) | $19.9_{\pm 2.4}$ | $36.2_{\pm 14.4}$ | $47.4_{\pm 18.1}$ | $52.5_{\pm 22.3}$ | $64.2_{\pm 12.6}$ | $68.4_{\pm 2.3}$ | $67.1_{\pm 2.7}$ | $65.2_{\pm 2.8}$ | $67.3_{\pm 3.4}$ | - |
| Buffer-and-Reinforce (Ours) | $8.7_{\pm 1.0}$ | $\mathbf{8.1}_{\pm 0.3}$ | $8.7_{\pm 0.3}$ | $\mathbf{8.2}_{\pm 0.3}$ | $\mathbf{8.8}_{\pm 1.2}$ | $76.0_{\pm 1.3}$ | $\mathbf{76.6}_{\pm 1.3}$ | $\mathbf{75.8}_{\pm 0.8}$ | $\mathbf{75.2}_{\pm 0.4}$ | - |

*Table 2.* **Comparison of safety and utility across varying user data sizes ($n$), ranging from 500 to 2,500.** Our framework consistently achieves the lowest Harmful Score while maintaining the highest Fine-tuning Accuracy, outperforming all baseline methods.

| Methods | Harmful Score (↓) | | | | | | Fine-tuning Accuracy (↑) | | | | | |
| --- | --- | --- | --- | --- | --- | --- | --- | --- | --- | --- | --- | --- |
| | n=500 | n=1000 | n=1500 | n=2000 | n=2500 | Average | n=500 | n=1000 | n=1500 | n=2000 | n=2500 | Average |
| SFT | 61.8 | 75.2 | 77.0 | 79.3 | 80.0 | 74.7 | 67.2 | 68.4 | 70.3 | 71.0 | 71.4 | 69.7 |
| LDIFS (Mukhoti et al., 2024) | 19.1 | 17.2 | 18.4 | 16.5 | 18.7 | 18.0 | 68.6 | 73.8 | 71.2 | 66.7 | 69.1 | 69.9 |
| SafeInstruct (Bianchi et al., 2024) | 29.0 | 21.5 | 19.0 | 19.4 | 19.0 | 21.6 | 68.9 | 69.8 | 70.0 | 70.6 | 70.4 | 69.9 |
| Lisa (Huang et al., 2024a) | 25.5 | 26.8 | 26.0 | 24.7 | 23.1 | 25.2 | 68.2 | 69.1 | 69.7 | 70.1 | 70.0 | 69.4 |
| Security Vector (Zhou et al., 2024) | 20.5 | 21.7 | 23.6 | 21.0 | 23.2 | 22.0 | 71.2 | 72.2 | 71.5 | 71.5 | 70.2 | 71.3 |
| AsFT (Yang et al., 2026) | 23.7 | 26.0 | 28.8 | 29.7 | 33.2 | 28.3 | 70.2 | 68.3 | 67.8 | 70.3 | 68.2 | 69.0 |
| SafeLoRA (Hsu et al., 2024) | 21.7 | 26.0 | 25.9 | 24.8 | 27.6 | 25.2 | 73.8 | 72.8 | 74.5 | 74.9 | 74.2 | 74.0 |
| Antidote (Huang et al., 2025a) | 27.4 | 25.3 | 45.6 | 50.4 | 56.7 | 41.1 | 73.6 | 74.7 | 74.7 | 75.9 | 75.7 | 74.9 |
| Panacea (Wang et al., 2026) | 35.7 | 41.0 | 81.9 | 88.3 | 62.9 | 62.0 | 49.7 | 65.5 | 55.1 | 63.2 | 62.2 | 59.1 |
| Buffer-and-Reinforce (Ours) | **8.5** | **8.4** | **8.2** | **8.7** | **9.1** | **8.6** | **75.1** | **77.5** | **77.7** | **76.3** | **76.7** | **76.7** |

# 6. Experiments

The goal of our experiments is to evaluate our framework on safety-alignment and user-task performance. To this end, we vary the contamination ratio of harmful prompts in user data, the number of user data samples, the target task (GSM8K (Cobbe et al., 2021), SST2 (Socher et al., 2013), AGNEWS (Zhang et al., 2015)), and the underlying base model (LLaMA3-8B-Instruct (Grattafiori et al., 2024), Gemma2-9B-it (Team et al., 2024), Qwen3-4B-Instruct-2507 (Yang et al., 2025)). By default, we use LLaMA3-8B-Instruct with 1,000 user data samples, a harmful ratio of 0.1, and GSM8K as the benign task, unless stated otherwise.

**Datasets.** Our experiments follow a cross-dataset setting that separates FaaS provider-side data from user-side data. The provider has access to 5,000 harmful queries with harmful and refusal responses from LAT (Sheshadri et al., 2024) and 5,000 benign queries with helpful responses from Alpaca (Taori et al., 2023). User data is constructed by mixing BeaverTails (Ji et al., 2023) harmful prompts with user-task samples according to a specified harmful data ratio.

**Metrics.** Following prior work (Huang et al., 2024b;a; 2025b;a), we evaluate fine-tuned models in terms of safety and utility. Safety is measured by the Harmful Score (HS), defined as the fraction of outputs on the BeaverTails test set that are identified as harmful by Beaver-Dam-7B (Ji et al., 2023). Utility is measured by Fine-tuning Accuracy (FA) on downstream benchmarks, including GSM8K, SST2, and AGNEWS. All metrics are measured after user fine-tuning.

**Baselines.** We assume that FaaS operates on already safety-aligned LLMs. Accordingly, alignment-stage methods that search for alternative safety-aligned weights are difficult to apply to pre-aligned LLMs and are therefore excluded from the main tables; we evaluate them separately in Appendix B.6. We primarily compare our fine-tuning framework against fine-tuning-stage defenses that can be directly applied to safety-aligned LLMs, including LDIFS (Mukhoti et al., 2024), SafeInstruct (Bianchi et al., 2024), Lisa (Huang et al., 2024a), Security Vector (Zhou et al., 2024), and AsFT (Yang et al., 2026), as well as post-fine-tuning-stage approaches such as SafeLoRA (Hsu et al., 2024), Antidote (Huang et al., 2025a), and Panacea (Wang et al., 2026). We also include standard supervised fine-tuning (SFT) as a reference baseline without defense. Details are provided in Appendix A.2.

## 6.1. Experiment Results

**Varying Harmful Ratio.** We evaluate the robustness of our Buffer-and-Reinforce fine-tuning framework under varying the proportion of harmful data in the user data by mea-

*Table 3.* **Comparison of safety and utility across different downstream tasks.** Our framework consistently yields low HS while maintaining the highest average FA across tasks, demonstrating strong safety without substantial degradation in task performance.

| Methods | GSM8K | | SST2 | | AGNEWS | | Average | |
|---|---|---|---|---|---|---|---|---|
| | HS ↓ | FA ↑ | HS ↓ | FA ↑ | HS ↓ | FA ↑ | HS ↓ | FA ↑ |
| SFT | 75.2 | 68.4 | 79.4 | **95.8** | 78.2 | 87.8 | 77.6 | 84.0 |
| LDIFS (Mukhoti et al., 2024) | 17.2 | 73.8 | 15.0 | 92.6 | 16.3 | 72.5 | 16.2 | 79.6 |
| SafeInstruct (Bianchi et al., 2024) | 21.5 | 69.8 | 17.6 | 95.0 | 16.1 | 86.6 | 18.4 | 83.8 |
| Lisa (Huang et al., 2024a) | 26.8 | 69.1 | 27.8 | 94.6 | 39.5 | 82.9 | 31.4 | 82.2 |
| Security Vector (Zhou et al., 2024) | 21.7 | 72.2 | 23.6 | 95.3 | 17.9 | 86.1 | 21.1 | 84.5 |
| AsFT (Yang et al., 2026) | 26.0 | 68.3 | 56.4 | 94.7 | 24.0 | 81.0 | 35.5 | 81.3 |
| SafeLoRA (Hsu et al., 2024) | 26.0 | 72.8 | 22.2 | 86.8 | 31.8 | 79.8 | 26.7 | 79.8 |
| Antidote (Huang et al., 2025a) | 25.3 | 74.7 | 28.3 | 93.5 | 28.8 | 82.3 | 27.5 | 83.5 |
| Panacea (Wang et al., 2026) | 41.0 | 65.5 | 73.4 | 89.8 | 53.7 | 86.6 | 56.0 | 80.6 |
| Buffer-and-Reinforce (Ours) | **8.4** | **77.5** | **10.2** | 93.8 | **7.5** | **88.0** | **8.7** | **86.4** |

*Table 4.* **Comparison of safety and utility across different base model architectures.** We report Harmful Score (HS) and Fine-tuning Accuracy (FA) on three instruction-tuned, safety-aligned LLMs: LLaMA3-8B-Instruct, Gemma2-9B-it, and Qwen3-4B-Instruct, along with their averages. Overall, our framework exhibits strong robustness across different model architectures.

| Methods | LLaMA3-8B-Instruct | | Gemma2-9B-it | | Qwen3-4B-Instruct | | Average | |
|---|---|---|---|---|---|---|---|---|
| | HS ↓ | FA ↑ | HS ↓ | FA ↑ | HS ↓ | FA ↑ | HS ↓ | FA ↑ |
| SFT | 75.2 | 68.4 | 59.1 | 80.0 | 72.2 | 79.6 | 68.8 | 76.0 |
| LDIFS (Mukhoti et al., 2024) | 17.2 | 73.8 | 4.1 | 81.0 | 71.0 | 79.9 | 30.8 | 78.2 |
| SafeInstruct (Bianchi et al., 2024) | 21.5 | 69.8 | 18.5 | 79.9 | 27.3 | 80.0 | 22.4 | 76.6 |
| Lisa (Huang et al., 2024a) | 26.8 | 69.1 | 7.7 | 79.7 | 22.6 | 80.9 | 19.0 | 76.6 |
| Security Vector (Zhou et al., 2024) | 21.7 | 72.2 | 6.5 | 81.2 | 16.9 | **81.1** | 15.0 | 78.2 |
| AsFT (Yang et al., 2026) | 26.0 | 68.3 | 5.5 | 77.4 | 23.5 | 58.0 | 18.3 | 67.9 |
| SafeLoRA (Hsu et al., 2024) | 26.0 | 72.8 | 7.8 | 78.8 | 14.2 | 56.8 | 20.7 | 79.4 |
| Antidote (Huang et al., 2025a) | 25.3 | 74.7 | 5.3 | 81.0 | 13.0 | 76.5 | 14.5 | 77.4 |
| Panacea (Wang et al., 2026) | 41.0 | 65.5 | 32.0 | 78.7 | 25.9 | 80.3 | 33.0 | 74.8 |
| Buffer-and-Reinforce (Ours) | **8.4** | **77.5** | **3.0** | **81.4** | **5.5** | 80.0 | **5.2** | **78.3** |

*Table 5.* **Computational cost comparison across baselines and our framework.** We report GPUTime (total run time) and GPUMemory (peak GPU memory usage), measured on A100 GPU, for each stage: Before Fine-tuning, User Fine-tuning, and Post-fine-tuning. Although our method requires preparing BufferLoRA and ReinforceLoRA, it incurs the same computational cost as SFT during user fine-tuning.

| Methods | Before Fine-tuning | | User Fine-tuning | | Post-fine-tuning | |
|---|---|---|---|---|---|---|
| | GPUTime (min) | GPUMemory (GB) | GPUTime (min) | GPUMemory (GB) | GPUTime (min) | GPUMemory (GB) |
| SFT | 0.00 | 0.00 | 3.93 | 39.11 | 0.00 | 0.00 |
| LDIFS (Mukhoti et al., 2024) | 0.00 | 0.00 | 7.24 | 55.53 | 0.00 | 0.00 |
| SafeInstruct (Bianchi et al., 2024) | 0.00 | 0.00 | 4.21 | 39.47 | 0.00 | 0.00 |
| Lisa (Huang et al., 2024a) | 0.00 | 0.00 | 11.42 | 40.86 | 0.00 | 0.00 |
| Security Vector (Zhou et al., 2024) | 38.29 | 69.93 | 3.93 | 39.11 | 0.00 | 0.00 |
| AsFT (Yang et al., 2026) | 1.71 | 14.14 | 202.92 | 119.10 | 0.00 | 0.00 |
| SafeLoRA (Hsu et al., 2024) | 0.00 | 0.00 | 3.93 | 39.11 | 0.89 | 34.03 |
| Antidote (Huang et al., 2025a) | 0.00 | 0.00 | 3.93 | 39.11 | 2.11 | 33.16 |
| Panacea (Wang et al., 2026) | 0.00 | 0.00 | 18.19 | 74.97 | $1.92e^{-4}$ | 76.04 |
| Buffer-and-Reinforce (Ours) | 30.04 | 41.10 | 3.93 | 39.11 | 0.25 | 26.00 |

suring HS and FA. As shown in Table 1, increasing the harmful ratio $p$ generally leads to higher HS and lower FA across most methods. All baselines exhibit substantially better safety than naive SFT, indicating the effectiveness of existing defenses to some extent. However, when $p \geq 0.3$, many baselines suffer from a sharp increase in HS, revealing limited robustness under heavily contaminated user data. In contrast, our method consistently achieves low HS and high FA even when $p \geq 0.3$. This result demonstrates that our framework can effectively preserve safety and utility even under challenging adversarial settings where users intentionally provide a large fraction of harmful data.

**Varying User Data Size.** We analyze the effect of user data size on safety and utility by varying the number of user-provided samples. Table 2 shows that both HS and FA generally increase as the dataset size $n$ grows, although the increase is not strictly proportional. This trend arises because the harmful ratio $p$ is fixed, increasing $n$ also increases the absolute number of harmful samples in the user-provided data. Despite these changes in harmful data scale, our framework yields the lowest HS and the highest FA across all values of $n$. These results demonstrate that our method remains robust to variations in user data size, effectively preserving both safety and downstream task performance.

**Diverse Downstream Tasks.** To demonstrate that our approach is effective beyond the default GSM8K setting, we further evaluate our framework on a broader range of downstream tasks, including SST2 and AGNEWS. These datasets differ substantially in domain and task characteristics, enabling a comprehensive assessment of our method's generality across both reasoning-oriented and classification-based workloads. As shown in Table 3, our method consistently achieves low HS and high FA on both SST2 and AGNEWS. These results indicate that the advantages of our framework are not limited to a specific task or dataset, but consistently generalize across diverse downstream applications.

**Diverse Model Architectures.** While our experiments primarily focus on LLaMA3-8B-Instruct as the default backbone, we further evaluate the effectiveness of our framework across different model architectures and parameter scales. Specifically, we conduct experiments on Gemma2-9B-it and Qwen3-4B-Instruct, which differ in both model size and architectural design. Table 4 shows that our framework exhibits strong safety and utility across these diverse architectures, achieving low HS while maintaining high FA. These results indicate that the benefits of our approach are not tied to a specific backbone model, but generalize across LLMs with varying capacities and architectural characteristics.

### 6.2. Computational Cost.

We compare the computational cost of our framework with baselines by measuring total GPU time and peak GPU memory usage across three stages: Before Fine-tuning, User Fine-tuning, and Post-fine-tuning. **Before Fine-tuning** stage corresponds to service preparation steps performed prior to receiving user data. For AsFT, this stage includes identifying safety directions; for Security Vector, it includes training the Security Vector; and for our method, it involves training BufferLoRA and ReinforceLoRA. **User Fine-tuning** stage adapts the LLM to user data, where fine-tuning-stage methods operate, and post-fine-tuning-stage methods typically perform SFT. **Post-fine-tuning** stage includes post-processing after user fine-tuning, where post-fine-tuning-stage methods are applied; for our framework, this contains projecting ReinforceLoRA via QR decomposition and merging it with UserLoRA.

Although our framework requires pre-training BufferLoRA and ReinforceLoRA prior to FaaS deployment, these components are trained only once and can be reused across all user fine-tuning sessions. As shown in Table 5, our method incurs the same computational cost as SFT during user fine-tuning, achieving substantially lower per-user fine-tuning cost than baselines. These results demonstrate that our framework is a practical and efficient solution for FaaS.

*Table 6.* **Ablation of our fine-tuning framework components.**

| Row | BufferLoRA | ReinforceLoRA | QR | HS ↓ | FA ↑ |
|-----|------------|---------------|-----|------|------|
| 1 | X | X | X | 75.2 | 68.4 |
| 2 | O | X | X | 18.0 | 75.4 |
| 3 | O | O | X | 4.7 | 74.0 |
| 4 | X | O | X | 9.6 | 67.3 |
| 5 | X | O | O | 21.8 | 56.6 |
| 6 | O | O | O | **8.4** | **77.5** |

### 6.3. Analysis

**Ablation Study.** Table 6 shows an ablation study to analyze the contributions of the components in our Buffer-and-Reinforce framework: BufferLoRA, ReinforceLoRA, and QR decomposition-based merging strategy (QR).

Comparing Rows 1 and 2 shows that BufferLoRA substantially reduces the HS while improving FA. This is because BufferLoRA saturates safety-degrading gradients during user fine-tuning and mitigates gradient conflicts between harmful updates and downstream task learning, which in turn benefits task performance as well as safety. Next, comparing Rows 2 and 3 highlights the role of ReinforceLoRA. Incorporating ReinforceLoRA further decreases HS, confirming its effectiveness in reinforcing model safety. However, this comes at the cost of reduced FA, as naively merging ReinforceLoRA with UserLoRA introduces interference that degrades user-task representations. This trade-off motivates the QR decomposition-based merging strategy. As shown in the last row, integrating only the components of ReinforceLoRA that are orthogonal to the UserLoRA restores and even improves FA, while incurring a modest yet competitive increase in HS. This result suggests that QR-based orthogonalization reduces interference between safety reinforcement and downstream task learning.

In addition, Rows 4 and 5 examine configurations without BufferLoRA. While ReinforceLoRA alone improves safety, FA remains low because harmful information is encoded into UserLoRA during user fine-tuning. QR-based orthogonalization is also less effective in this setting, since harmful updates already embedded in UserLoRA can be treated as part of the user-task subspace and preserved during merging. Overall, these results show that BufferLoRA, ReinforceLoRA, and QR-based merging are complementary and jointly necessary for achieving strong safety and utility.

**Effect of Training Data Size for BufferLoRA and ReinforceLoRA.** We analyze how the number of training data used for BufferLoRA and ReinforceLoRA affects safety and utility after user fine-tuning. Specifically, we vary the number of samples used to train each module and evaluate HS and FA. As shown in Table 7, increasing the training data for both BufferLoRA and ReinforceLoRA reduces HS and improves FA. In particular, training BufferLoRA

*Table 7.* **Effect of training data size for BufferLoRA and Re-inforceLoRA.** Insufficient BufferLoRA training due to limited training data degrades safety after user fine-tuning, highlighting the importance of sufficiently strong jailbreaking of BufferLoRA.

| # Samples | BufferLoRA | ReinforceLoRA | ReinforceLoRA +UserLoRA | |
|---|---|---|---|---|
| | HS | HS | HS ↓ | FA ↑ |
| 100 | 23.8 | 9.6 | 20.8 | 75.7 |
| 500 | 84.1 | 7.1 | 11.0 | 76.1 |
| 1,000 | 89.0 | 6.2 | 10.3 | 76.1 |
| 3,000 | 86.8 | 3.8 | 9.2 | 77.4 |
| 5,000 | 88.2 | 4.3 | 8.4 | 77.5 |

with a sufficiently large set of harmful samples induces a strongly jailbroken state, enabling more effective and stable saturation of safety-degrading gradients during user fine-tuning. This strong jailbreaking is critical to the effectiveness of our framework. In contrast, when BufferLoRA is trained with limited harmful data, it fails to fully converge on harmful queries. In this case, even if ReinforceLoRA alone achieves low HS, the overall framework cannot maintain strong safety after user fine-tuning. These results indicate that strong jailbreaking via BufferLoRA is essential, and that ReinforceLoRA alone cannot compensate for insufficient BufferLoRA training.

## 7. Limitations

While the effectiveness of our framework depends on sufficiently strong jailbreaking of the base LLM, our framework relies on the ability to induce a temporarily jailbroken state through BufferLoRA. Although our experiments demonstrate its effectiveness across multiple models, this assumption may become less reliable as future models become more robust to jailbreaking In such cases, harmful loss saturation may be weaker, potentially reducing the buffering effect. Developing stronger or adaptive BufferLoRA training strategies is an important direction for future work.

We also evaluate our framework on models in the 4B-13B parameter range. While this covers several widely used open-weight LLMs, practical FaaS systems may involve substantially larger frontier models. Due to computational costs and limited access to frontier-scale model weights, we could not directly verify whether the same behavior holds at larger scales. Future work should examine the scalability of our framework on larger and more robust models.

## 8. Conclusion

In this paper, we introduced a new perspective on defending against harmful fine-tuning attacks in Fine-tuning-as-a-Service (FaaS). Unlike prior approaches that explicitly suppress harmful updates through regularization, we showed that a temporarily jailbroken LLM naturally neutralizes

safety-degrading gradients via harmful loss convergence, while preserving the ability to learn benign, user-task knowledge. Based on this observation, we proposed a Buffer-and-Reinforce fine-tuning framework that leverages a removable BufferLoRA to saturate harmful gradients during user fine-tuning, a ReinforceLoRA to reinforce safety, and a QR decomposition-based merging strategy to resolve conflicts between safety reinforcement and user-task learning. Extensive experiments across varying harmful data ratios, user data sizes, downstream tasks, and model architectures demonstrated that our framework consistently achieves strong safety and high utility, outperforming existing fine-tuning-stage and post-fine-tuning-stage defenses. Notably, during user fine-tuning, our method requires no additional safety-alignment data and incurs the same computational cost as SFT, making it practical solution for real-world FaaS.

## Impact Statement

This work aims to improve the safety of personalized LLMs in Fine-tuning-as-a-Service (FaaS). By buffering harmful updates during user fine-tuning and reinforcing safety after adaptation, our framework can help providers customize models while reducing safety degradation caused by malicious or contaminated user data. This supports safer deployment of LLM personalization services in practical settings.

However, our framework intentionally relies on a temporarily jailbroken state. Although BufferLoRA is frozen during user fine-tuning and removed before deployment, improper access to such modules could introduce misuse risks. Therefore, BufferLoRA should remain under secure provider-side control, with appropriate access restrictions. Overall, this work contributes to safer LLM personalization while emphasizing the need for careful governance of defense mechanisms based on controlled harmful-behavior activation.

## Acknowledgements

This work was supported by Institute of Information & Communication Technology Planning & Evaluation (IITP) grant funded by the Korea government (MSIT) (No. RS-2025-02215344, Development of AI Technology with Robust and Flexible Resilience Against Risk Factors).

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

# A. Experimental Details

## A.1. Training Setup

Across all experiments, we use a unified LoRA (Hu et al., 2022) configuration with rank 32 and scaling factor $\alpha = 64$, and apply LoRA to the q_proj, v_proj, gate_proj, up_proj, and down_proj modules. For LLaMA3-8B-Instruct (Grattafiori et al., 2024), we use a learning rate of $1 \times 10^{-5}$ and train for 3 epochs on GSM8K (Cobbe et al., 2021) and AGNEWS (Zhang et al., 2015), while a higher learning rate of $1 \times 10^{-4}$ is used for SST2 (Socher et al., 2013). For Gemma2-9B-it (Team et al., 2024), we adopt different learning rates for different LoRA modules to ensure effective jailbreaking and stable fine-tuning: Buffer-LoRA and ReinforceLoRA are trained with a learning rate of $5 \times 10^{-5}$ for 3 epochs, while UserLoRA is trained with a smaller learning rate of $5 \times 10^{-6}$ for 3 epochs. For Qwen3-4B-Instruct-2507 (Yang et al., 2025), we use a learning rate of $5 \times 10^{-5}$ and train for 2 epochs. All experiments are conducted on four RTX 3090 GPUs, except for experiments measuring computational cost. For evaluation, we follow prior works (Huang et al., 2025a; 2024a; 2025b; 2024b) and report results on 1,000 samples from the GSM8K test set, 872 samples from the SST2 validation set, and 1,000 samples from the AGNEWS test set.

## A.2. Details on Baselines

Prior work on defending harmful fine-tuning attacks in Fine-tuning-as-a-Service (FaaS) spans alignment-stage, fine-tuning-stage, and post-fine-tuning-stage methods, depending on when the defense is applied. Following the experimental setting described in the main manuscript, we focus on fine-tuning-stage and post-fine-tuning-stage methods that are directly applicable to already safety-aligned LLMs. Below, we provide detailed descriptions and implementation settings for all evaluated baselines. All hyperparameter search results are reported in Table A1.

**SFT (Supervised Fine-Tuning)**
- **User Fine-tuning:** Applying standard supervised fine-tuning to the safety-aligned LLM using user-provided data.

**LDIFS (Mukhoti et al., 2024)**
- **User Fine-tuning:** Mitigating harmful updates through a regularization term that penalizes deviations between the representations of the original and fine-tuned models; we set $\lambda = 0.01$, which yielded the best performance in our experimental setting.

**SafeInstruct (Bianchi et al., 2024)**
- **User Fine-tuning:** Fine-tuning a safety-aligned LLM using a combined dataset consisting of user data and additional safety-alignment data samples. Following prior work (Bianchi et al., 2024), when the user dataset contains 1,000 samples, we augment it with 100 safety-alignment data samples (10%).

**Lisa (Huang et al., 2024a)**
- **User Fine-tuning:** Alternates optimization steps between safety-alignment data and user data with a regularization objective. We set the regularization strength to $\rho = 10$ and adopt a 10/90 split between alignment and fine-tuning steps, which achieves the best performance in our experimental setting.

**Security Vector (Zhou et al., 2024)**
- **Before Fine-tuning:** Training separable Security Vector on a small set of harmful samples, which make the model exhibit harmful behavior when activated. Following the original paper, we use a KL loss with weight 0.01 to preserve the model's behavior on benign data.
- **User Fine-tuning:** Activating the security vectors during supervised fine-tuning on user-provided data.

**AsFT (Yang et al., 2026)**
- **Before Fine-tuning:** Extracting safety directions by computing the difference between a safety-aligned LLM and its corresponding base model, following SafeLoRA (Hsu et al., 2024).
- **User Fine-tuning:** Defining harmful directions as those orthogonal to the extracted safety directions and suppressing updates along these directions via regularization. We set the regularization strength to $\lambda = 10$, which yields the best performance in our experimental setting.

**SafeLoRA (Hsu et al., 2024)**
- **User Fine-tuning:** Applying standard supervised fine-tuning to a safety-aligned LLM using user-provided data. (equal to SFT)
- **Post-fine-tuning:** Obtaining safety directions from the parameter difference between a safety-aligned LLM and the corresponding base model, and projecting the fine-tuned LLM onto these safety directions to restore safety. The original weights are replaced with their projected weights when the cosine similarity between them falls below a threshold, which we set to 0.6 based on our experiments.

**Antidote (Huang et al., 2025a)**
- **User Fine-tuning:** Applying standard supervised fine-tuning to a safety-aligned LLM using user-provided data. (equal to SFT)
- **Post-fine-tuning:** Applying neuron masking based on Wanda scoring (Sun et al., 2024) to suppress harmful

*Table A1.* **Hyperparameter search for baseline methods.** To ensure fair comparison and assess the sensitivity of each baseline to its hyperparameters, we empirically explore key hyperparameters for all baseline methods. The optimal configurations, selected based on the best trade-off between safety and utility, are highlighted in **bold** and used in the main experiments.

| *(a)* LDIFS | | | | *(b)* Lisa | | | | *(c)* AsFT | | |
|---|---|---|---|---|---|---|---|---|---|---|
| $\rho$ | HS $\downarrow$ | FA $\uparrow$ | | $\rho$ | HS $\downarrow$ | FA $\uparrow$ | | $\lambda$ | HS $\downarrow$ | FA $\uparrow$ |
| 0.001 | 18.3 | 74.3 | | 0.01 | 73.6 | 66.9 | | 1.0 | 38.0 | 63.5 |
| **0.01** | **17.2** | **73.8** | | 0.1 | 73.3 | 67.4 | | 2.0 | 33.5 | 65.5 |
| 0.1 | 29.7 | 73.9 | | 1.0 | 55.8 | 69.1 | | 5.0 | 29.7 | 67.1 |
| 1.0 | 74.6 | 69.1 | | **10.0** | **26.8** | **69.1** | | **10.0** | **26.0** | **68.3** |
| 10.0 | 71.0 | 69.4 | | 20.0 | 28.9 | 70.8 | | 20.0 | 25.1 | 67.8 |

| *(d)* SafeLoRA | | | | *(e)* Antidote | | | | *(f)* Panacea | | | |
|---|---|---|---|---|---|---|---|---|---|---|---|
| Threshold | HS $\downarrow$ | FA $\uparrow$ | | Dense Ratio | HS $\downarrow$ | FA $\uparrow$ | | $\epsilon_\rho$ | $\lambda$ | HS $\downarrow$ | FA $\uparrow$ |
| 0.5 | 29.9 | 73.0 | | 0.01 | 20.6 | 72.1 | | 1.0 | 0.001 | 43.8 | 64.4 |
| **0.6** | **26.0** | **72.8** | | **0.05** | **25.3** | **74.7** | | 2.0 | 0.001 | 48.3 | 37.3 |
| 0.7 | 21.6 | 69.3 | | 0.1 | 40.2 | 72.4 | | **1.0** | **0.0001** | **41.0** | **65.5** |

neurons, using a masking ratio of $\alpha = 0.05$ that yields the best performance in our setting.

**Panacea (Wang et al., 2026)**

- **User Fine-tuning:** Optimizing an adaptive, norm-bounded parameter perturbation to increase harmful loss, where the perturbation bound is set to $\epsilon_\rho = 1$, which yields the strongest safety in our experiments.

- **Post-fine-tuning:** Applying the optimized perturbation to the fine-tuned model to restore safety-alignment while preserving downstream task performance; we set the trade-off parameter to $\lambda = 10^{-4}$ based on our experiments.

### A.3. Hyperparameter Configuration and Sensitivity Analysis

We analyze the sensitivity of our framework to the training data size and number of epochs for BufferLoRA, ReinforceLoRA, and UserLoRA. Table A2 reports the Harmful Score (HS) and Fine-tuning Accuracy (FA) results under several settings. By default, we train BufferLoRA on 5,000 harmful samples for 3 epochs, ReinforceLoRA on 5,000 safety-alignment samples consisting of 2,500 harmful and 2,500 benign samples for 3 epochs, and UserLoRA on 1,000 user samples for 3 epochs.

Overall, the framework is robust to moderate hyperparameter changes. Changing the data size of BufferLoRA or ReinforceLoRA from 5,000 to 3,000 or 10,000 causes only small changes in the final harmful score and fine-tuning accuracy. Also, changing the number of epochs from 1 to 10 also has limited impact, suggesting that the proposed safety mechanism remains effective across different degrees of user adaptation. We analyze the effect of the user data

size on UserLoRA in Table 2.

In practice, BufferLoRA and ReinforceLoRA should be configured by monitoring harmful score, since they directly control temporary jailbreaking and safety reinforcement. Whereas, UserLoRA is task-dependent, and its configuration should be selected according to the type and amount of user data.

## B. Additional Analysis and Experiments

### B.1. Equivalence of Projection Subspaces for LoRA

For soft orthogonalization in the post-fine-tuning stage, we require an efficient method to project the ReinforceLoRA $W_R$ onto the orthogonal complement of the UserLoRA output subspace. Rather than operating directly on the full matrix $W_U \in \mathbb{R}^{d_{\text{out}} \times d_{\text{in}}}$, we exploit the low-rank structure of LoRA to significantly reduce computational complexity. Below, we show that orthogonal projection with respect to $W_U$ can be equivalently performed using only its decoder matrix $B_U$.

**Proposition B.1.** *(Equivalence of Projection Subspaces) Let the UserLoRA be defined as $W_U = B_U A_U$, where $B_U \in \mathbb{R}^{d_{out} \times r}$ and $A_U \in \mathbb{R}^{r \times d_{in}}$ with $r \ll \min(d_{out}, d_{in})$. Assuming that $A_U$ has full row rank, the orthogonal projection onto the column space of $W_U$ is equivalent to the orthogonal projection onto the column space of $B_U$.*

*Proof.* The column space of a matrix $W_U$, denoted by $\mathcal{C}(W_U)$, is defined as

$$\mathcal{C}(W_U) = \{W_U v \mid v \in \mathbb{R}^{d_{\text{in}}}\}.$$

*Table A2.* Hyperparameter sensitivity of BufferLoRA, ReinforceLoRA, and UserLoRA.

| Module | BufferLoRA Data Size | Epochs | ReinforceLoRA Data Size | Epochs | UserLoRA Data Size | Epochs | BufferLoRA HS | ReinforceLoRA HS | Final HS | FA |
|--------|------|--------|------|--------|------|--------|------|------|------|------|
| Default | 5K | 3 | 5K | 3 | 1K | 3 | 88.2 | 4.3 | 8.4 | 77.5 |
| BufferLoRA | 5K | 1 | 5K | 3 | 1K | 3 | 87.9 | 3.7 | 10.6 | 76.6 |
| | 5K | 10 | 5K | 3 | 1K | 3 | 87.5 | 6.2 | 11.7 | 69.2 |
| | 3K | 3 | 5K | 3 | 1K | 3 | 86.8 | 3.3 | 7.8 | 77.1 |
| | 10K | 3 | 5K | 3 | 1K | 3 | 87.4 | 3.9 | 8.2 | 76.0 |
| ReinforceLoRA | 5K | 3 | 5K | 1 | 1K | 3 | 88.2 | 4.9 | 8.2 | 74.7 |
| | 5K | 3 | 5K | 10 | 1K | 3 | 88.2 | 4.6 | 9.9 | 75.2 |
| | 5K | 3 | 3K | 3 | 1K | 3 | 88.2 | 4.9 | 7.4 | 74.8 |
| | 5K | 3 | 10K | 3 | 1K | 3 | 88.2 | 3.9 | 8.7 | 76.6 |
| UserLoRA | 5K | 3 | 5K | 3 | 1K | 1 | 88.2 | 4.3 | 8.5 | 76.9 |
| | 5K | 3 | 5K | 3 | 1K | 10 | 88.2 | 4.3 | 9.3 | 77.9 |

**Inclusion ($\subseteq$).** For any vector $y \in \mathcal{C}(W_U)$, there exists $x \in \mathbb{R}^{d_{\text{in}}}$ such that

$$y = W_U x = (B_U A_U)x = B_U(A_U x).$$

Let $z = A_U x$. Then $y = B_U z$, which implies $y \in \mathcal{C}(B_U)$. Hence,

$$\mathcal{C}(W_U) \subseteq \mathcal{C}(B_U).$$

**Reverse inclusion ($\supseteq$).** Since $A_U \in \mathbb{R}^{r \times d_{\text{in}}}$ has full row rank, the linear map $f(x) = A_U x$ is surjective onto $\mathbb{R}^r$. Therefore, for any $z \in \mathbb{R}^r$, there exists $x \in \mathbb{R}^{d_{\text{in}}}$ such that $A_U x = z$. Consequently, any vector $B_U z \in \mathcal{C}(B_U)$ can be written as

$$B_U z = B_U(A_U x) = W_U x,$$

which implies

$$\mathcal{C}(B_U) \subseteq \mathcal{C}(W_U).$$

Combining both inclusions yields

$$\mathcal{C}(W_U) = \mathcal{C}(B_U).$$

$\square$

The equivalence in Proposition B.1 relies on the assumption that $A_U$ has full row rank. In practice, we empirically verify that this assumption holds for trained UserLoRA. Specifically, we measure the effective rank of $W_U$ and observe that it is equal to the full rank $r$ across layers, indicating that UserLoRA utilizes its full representational capacity without rank collapse. As a result, the approximation $\mathcal{C}(W_U) \approx \mathcal{C}(B_U)$ is well justified in practice, supporting the use of projection based on $B_U$.

Since the column spaces of $W_U$ and $B_U$ are identical, their corresponding orthogonal projection operators are also identical. Let $P_{W_U}$ and $P_{B_U}$ denote the orthogonal projectors onto $\mathcal{C}(W_U)$ and $\mathcal{C}(B_U)$, respectively. Then, $P_{W_U} = P_{B_U}$.

As a result, projecting onto the column space of $W_U$ can be performed using only $B_U$. In practice, we compute this projector via the QR decomposition $B_U = QR$, yielding $P_{B_U} = QQ^\top$.

### B.2. Equivalence of Principal Subspace Restriction

In the post-fine-tuning stage, we remove ReinforceLoRA components that interfere with the user task by projecting them onto the orthogonal complement of the output subspace induced by UserLoRA. While Section B.1 assumes that UserLoRA is full rank, this assumption may be violated due to rank collapse in rare cases. In such cases, directly operating with the nominal rank can lead to excessive removal of ReinforceLoRA components. To address this issue, we identify an effective UserLoRA subspace by performing an eigen-decomposition of the Gram matrix $G = A_U A_U^\top$ and restricting the projection to this subspace. This section provides a formal justification that this eigenvector-based restriction recovers the task-relevant output subspace induced by UserLoRA.

**Proposition B.2.** *(Equivalence to Principal Subspace Restriction) Let the UserLoRA be $W_U = B_U A_U$, where $B_U \in \mathbb{R}^{d_{out} \times r}$ and $A_U \in \mathbb{R}^{r \times d_{in}}$. Let $G = A_U A_U^\top$ be the Gram matrix of the input factor. Let $\tilde{A}_U$ denote the rank-$k$ truncation of $A_U$ obtained by singular value decomposition, and define the induced weight $\tilde{W}_U = B_U \tilde{A}_U$. Restricting $B_U$ to the subspace spanned by the top-$k$ eigenvectors of $G$ yields a basis whose span equals the column space of $\tilde{W}_U$, up to degenerate (zero singular value) directions.*

*Proof.* We begin with the Singular Value Decomposition (SVD) of the input factor:

$$A_U = U_A \Sigma_A V_A^\top, \tag{10}$$

where $U_A \in \mathbb{R}^{r \times r}$ contains the left singular vectors and $\Sigma_A = \text{diag}(\sigma_i)$ the singular values.

The Gram matrix can be written as

$$G = A_U A_U^\top = U_A \Sigma_A^2 U_A^\top. \tag{11}$$

Since $G$ is symmetric positive semi-definite, its eigenvectors span the same subspaces as the left singular vectors of $A_U$, and its eigenvalues satisfy $\lambda_i = \sigma_i^2$.

Let $U_{A,k}$ denote the matrix of the top-$k$ left singular vectors of $A_U$. Selecting the top-$k$ eigenvectors of $G$ therefore yields a matrix $V_{\text{eff}}$ whose columns span the same subspace as $U_{A,k}$ (up to orthogonal transformations within degenerate eigenspaces).

Consider the rank-$k$ truncation of $A_U$,

$$\tilde{A}_U = U_{A,k} \Sigma_{A,k} V_{A,k}^\top, \qquad (12)$$

which induces the effective update

$$\tilde{W}_U = B_U \tilde{A}_U. \qquad (13)$$

The column space of this update is given by

$$\mathcal{C}(\tilde{W}_U) = \mathcal{C}(B_U U_{A,k}), \qquad (14)$$

up to directions corresponding to zero singular values.

Defining $\hat{B} = B_U V_{\text{eff}}$, and using the fact that

$$\mathcal{C}(V_{\text{eff}}) = \mathcal{C}(U_{A,k}), \qquad (15)$$

we obtain

$$\mathcal{C}(\hat{B}) = \mathcal{C}(B_U V_{\text{eff}}) = \mathcal{C}(B_U U_{A,k}) = \mathcal{C}(\tilde{W}_U). \qquad (16)$$

Applying QR decomposition to $\hat{B}$ therefore yields an orthonormal basis $Q_B$ that spans exactly the output subspace induced by the dominant components of UserLoRA. $\square$

The proposition shows that the eigenvector-based restriction used in our method recovers the task-relevant output subspace of UserLoRA under rank collapse cases. As a result, projecting ReinforceLoRA onto the orthogonal complement of $Q_B$ removes interference strictly with respect to the user's effective task trajectory, while avoiding unnecessary suppression caused by collapsed or noisy dimensions.

### B.3. Limited Effectiveness in Mitigating Harmful Gradients of Prior Work

Beyond analyzing the Safety Gradient Score of safety-aligned LLMs, we further examine whether prior defense methods effectively mitigate safety-degrading gradients under fine-tuning on harmful (Ji et al., 2023) and harmless data (Cobbe et al., 2021). To this end, we compare the Safety Gradient Scores of safety-aligned LLMs with prior defenses against those of a jailbroken LLM. Notably, the Safety Gradient Score is measured without training and depends only on the initial model parameters and the loss function. As a result, methods that alternate between user fine-tuning

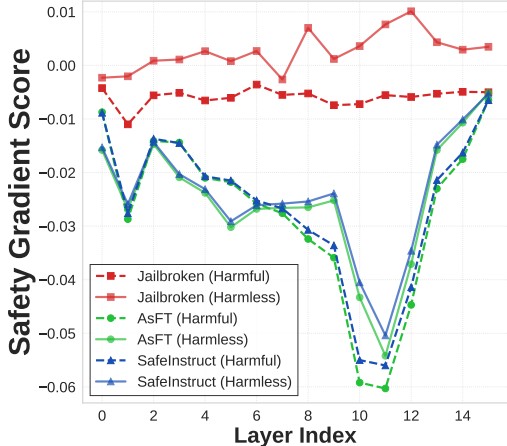

*Figure A1.* **Comparison of Safety Gradient Scores for prior defense methods and a jailbroken LLM.** The plot demonstrates that, despite defensive mechanisms, both SafeInstruct and AsFT exhibit negative Safety Gradient Scores across layers on harmful and harmless data. This indicates their limited effectiveness in mitigating safety-degrading gradient updates during fine-tuning.

and alignment tuning, such as LDIFS (Mukhoti et al., 2024) and Lisa (Huang et al., 2024a), are not compatible with this analysis. We therefore focus on methods with fixed training objectives, specifically SafeInstruct (Bianchi et al., 2024) and AsFT (Yang et al., 2026).

Figure A1 shows that safety-related gradients from these prior methods exhibit negative Safety Gradient Scores on both harmful and harmless data, similar to those observed for standard supervised fine-tuning on safety-aligned LLMs. This indicates that, despite their defensive mechanisms, these approaches do not sufficiently reduce safety-degrading gradients during fine-tuning. These results highlight a limitation of prior methods in directly mitigating safety-degrading gradients. While such approaches may reduce harmful outputs at the response level, they appear less effective at addressing safety degradation at the gradient level, which is critical for robustness against harmful fine-tuning attacks.

### B.4. Effect of Strength of QR Decomposition Projection

We investigate the effect of the projection strength hyperparameter $\alpha$, which controls the degree of orthogonalization applied to ReinforceLoRA and the trade-off between model safety and downstream utility. Specifically, $\alpha$ determines how aggressively ReinforceLoRA components aligned with the user-task subspace are suppressed. Table A3 reports the HS and FA under different values of $\alpha$. When $\alpha = 0$, ReinforceLoRA is directly added to UserLoRA without projection. While this naive merging achieves the lowest HS, it substantially degrades task performance, indicating strong interference between safety and user-task learning.

*Table A3.* **Effect of projection strength $\alpha$ on safety and utility.** $\alpha = 0.1$ provides the best trade-off between HS and FA.

| $\alpha$ | HS ↓ | FA ↑ |
|---|---|---|
| 0.0 | 4.7 | 74.0 |
| **0.1** | **8.4** | **77.5** |
| 0.3 | 9.5 | 75.8 |
| 0.5 | 8.7 | 77.2 |
| 1.0 | 8.7 | 76.2 |

*Table A4.* **Different Harmful Datasets for BufferLoRA and ReinforceLoRA.**

| Dataset | BufferLoRA | ReinforceLoRA | ReinforceLoRA +UserLoRA | |
|---|---|---|---|---|
| | HS ↓ | HS ↓ | HS ↓ | FA ↑ |
| BeaverTails | 83.0 | 10.2 | 13.5 | 77.4 |
| AdvBench | 80.8 | 4.4 | 11.9 | 74.5 |
| SafeRLHF | 82.4 | 1.9 | 14.1 | 78.4 |
| LAT | 88.0 | 4.4 | 8.4 | 77.5 |

*Table A5.* **Comparison with Alignment-stage Solutions**

| Method | HS ↓ | FA ↑ |
|---|---|---|
| RepNoise (Rosati et al., 2024) | 57.2 | 37.9 |
| Vaccine (Huang et al., 2024b) | 60.9 | 27.4 |
| Booster (Huang et al., 2025b) | 58.1 | 40.0 |
| Buffer-and-Reinforce (ours) | 8.4 | 77.5 |

Introducing a mild projection with $\alpha = 0.1$ effectively preserves task performance, with only a slight increase in HS. Interestingly, increasing $\alpha$ further does not yield additional gains in utility and results in unstable safety behavior. These results suggest that a modest degree of projection is sufficient to mitigate interference without compromising core safety-alignment. Based on this trade-off, we set $\alpha = 0.1$ for all our experiments.

### B.5. Impact of Training Dataset Choice for BufferLoRA and ReinforceLoRA

In our main manuscript, BufferLoRA and ReinforceLoRA are trained using the LAT dataset (Sheshadri et al., 2024), and robustness is evaluated on user-provided harmful data from BeaverTails (Ji et al., 2023). To further examine whether our framework remains effective when the FaaS provider relies on different datasets, we additionally train BufferLoRA and ReinforceLoRA using other datasets that contain both harmful responses and refusal responses, including BeaverTails, AdvBench (Zou et al., 2023) [1], and PKU-Safe-RLHF (Ji et al., 2025).

As shown in Table A4, BufferLoRA and ReinforceLoRA trained on these datasets are still able to support learning user-specific tasks while reinforcing safety. However, their

---

[1] https://huggingface.co/datasets/Baidicoot/augmented_advbench_v3_filtered

performance is suboptimal compared to models trained using LAT. This difference arises because BufferLoRA trained on these datasets induces a weaker jailbroken state, which less effectively saturates safety-degrading gradients during user fine-tuning.

These findings indicate that while our framework is not restricted to a specific dataset and can leverage various sources of harmful data, selecting or constructing datasets that induce sufficiently strong jailbreaking is an important factor for achieving optimal performance in our framework.

### B.6. Comparison with Alignment-stage Solutions

In our main experiments, we focus on fine-tuning-stage and post-fine-tuning-stage solutions, as these approaches can be directly applied to safety-aligned LLMs and are therefore more practical in FaaS settings. Nevertheless, alignment-stage solutions have also been proposed as defenses against harmful fine-tuning attacks. For completeness, we additionally evaluate representative alignment-stage solutions on LLaMA3-8B and compare them with our approach.

As shown in Table A5, alignment-stage solutions exhibit substantially lower performance in terms of both HS and FA. This degradation is attributed to the cross-domain setting considered in our experiments. Specifically, we assume that the harmful data available to the FaaS provider and the harmful data provided by users follow different distributions. Alignment-stage solutions aim to construct safety-aligned weights that are robust to harmful fine-tuning with respect to the provider's harmful data. As a result, these methods are easy to overfit to the provider's harmful data distribution and may generalize poorly to user harmful data. Moreover, as noted in prior work (Ham et al., 2025), weights obtained through alignment-stage optimization are often suboptimal for downstream task learning. This limits their ability to achieve high utility when subsequently fine-tuned on user-specific tasks. Consequently, these factors explain the inferior HS and FA observed for alignment-stage solutions in our cross-domain evaluation and highlight their limitations in practical FaaS scenarios.

### B.7. Effectiveness on More Robust and Larger Models

Our framework assumes that BufferLoRA can induce temporary jailbreaking through harmful fine-tuning. This assumption is more feasible than standard inference-time jailbreak attacks, because BufferLoRA is trained in a white-box setting where model parameters can be directly updated. Thus, the goal is not to discover an input-level jailbreak prompt, but to train a removable adapter that temporarily weakens safety during user fine-tuning.

Table A6 evaluates this assumption on models with stronger safety-alignment and a larger model scale. Under the same

*Table A6.* **Results across models with strengthened safety-alignment and a larger 13B model.**

| Model | Method | BufferLoRA HS ↓ | ReinforceLoRA HS ↓ | Final HS ↓ | Final FA ↑ |
|-------|--------|-----------------|--------------------|-----------|-----------|
| LLaMA3-8B-Instruct | Base | – | – | 19.2 | 62.8 |
|  | SFT | – | – | 75.2 | 68.4 |
|  | Ours | 88.2 | 4.3 | **8.4** | **77.5** |
| LLaMA3-8B-LAT | Base | – | – | – | 73.5 |
|  | SFT | – | – | 74.0 | 70.6 |
|  | Ours | 87.6 | 2.1 | **3.1** | **75.2** |
| LLaMA3-8B-ReFAT | Base | – | – | 16.2 | 40.2 |
|  | SFT | – | – | 74.0 | 68.9 |
|  | Ours | 87.3 | 2.2 | **3.5** | **71.1** |
| LLaMA2-13B-Chat | Base | – | – | 6.4 | 28.8 |
|  | SFT | – | – | 33.2 | 33.1 |
|  | Ours | 85.7 | 1.5 | **4.3** | **35.4** |

*Table A7.* **Cross-attack generalization across jailbreak attacks.**

|  | Backdoor HS | Backdoor FA | GCG HS | PAIR HS | TAP HS | FA |
|--|-------------|-------------|--------|---------|--------|----|
| SFT | 73.9 | 68.7 | 74.7 | 68.4 | 53.0 | 68.4 |
| Ours | 8.8 | 76.1 | 3.1 | 27.0 | 20.0 | 76.1 |

*Table A8.* **Comparison with LoRA merging baselines.**

| Method | HS ↓ | FA ↑ |
|--------|------|------|
| TaskArithmetic | 12.4 | 70.9 |
| TIES-Merging | 7.9 | 74.8 |
| LoRA-LEGO | 6.4 | 74.2 |
| Buffer-and-Reinforce (ours) | 8.4 | 77.5 |

training budget of 3 epochs, 5,000 harmful samples, and 5,000 safety-alignment samples, our framework remains effective on LAT (Sheshadri et al., 2024) and ReFAT (Yu et al., 2025) models, whose safety has been further reinforced, as well as on LLaMA2-13B-Chat (Touvron et al., 2023). Across these settings, BufferLoRA still induces strong temporary jailbreaking, ReinforceLoRA achieves low harmful scores, and the final model maintains low harmful scores while improving fine-tuning accuracy.

These results indicate that our method is not specific to the default LLaMA3-8B-Instruct setting. Within the evaluated 4B-13B range, we do not observe a need to increase the harmful or safety-alignment data budget as model size or safety strength increases. However, this does not imply that scale or robustness is irrelevant in general. More robust frontier models may require larger budgets or stronger BufferLoRA training, and evaluating such models remains future work.

### B.8. Cross-Attack Generalization

Harmful fine-tuning with malicious query-response pairs is itself a strong jailbreak threat in the FaaS setting, since an attacker can directly modify model parameters by submitting harmful user data. However, to examine whether our method generalizes beyond this fine-tuning-based attack scenario, we further evaluate it against trigger-based (Gu et al., 2019) and advanced jailbreak attacks, including GCG (Zou et al., 2023), PAIR (Chao et al., 2025), and TAP (Mehrotra et al.,

2024). Here, trigger-based attacks induce harmful behavior through predefined backdoor triggers, GCG optimizes adversarial suffixes to elicit harmful responses, PAIR uses an attacker LLM to iteratively generate jailbreak prompts, and TAP extends this idea with tree-structured search over candidate prompts. Table A7 shows that our method remains robust across diverse jailbreak attacks. This indicates that the proposed framework does not merely suppress a narrow attack-specific pattern, but improves broader resistance to harmful behavior elicitation.

### B.9. Comparison with Representative LoRA Merging Methods

We compare our QR-based merging strategy with representative LoRA merging methods, including Task Arithmetic (Ilharco et al., 2023), TIES-Merging (Yadav et al., 2023), and LoRA-LEGO (Zhao et al., 2025). Table A8 shows that our method achieves the best overall safety-utility trade-off. Although some baselines obtain slightly lower harmful scores, they cause a clearer drop in fine-tuning accuracy.

This result is consistent with our motivation. In our setting, UserLoRA is trained under BufferLoRA-based fine-tuning and therefore contains limited harmful information. Thus, the key objective at the merging stage is to preserve the utility of UserLoRA while injecting the safety behavior of ReinforceLoRA with minimal interference. Generic LoRA merging methods do not explicitly account for this asymmetric priority between the two adapters. By contrast, our

*Table A9.* **Comparison between SafeLoRA and QR-based merging on Energy Retain, Energy Damage, Harmful Score (HS), and Fine-tuning Accuracy (FA).**

| Method | Energy Retain | Energy Damage | HS ↓ | FA ↑ |
|---|---|---|---|---|
| SafeLoRA (Hsu et al., 2024) | $1.20e^{-5}$ | 0.999 | 18.0 | 63.3 |
| Buffer-and-Reinforce (ours) | 1.09 | $1.31e^{-3}$ | 8.4 | 77.5 |

*Table A10.* **Results under larger user-data settings on GSM8K.**

| Methods | Harmful Score (↓) | | | | Fine-tuning Accuracy (↑) | | | |
|---|---|---|---|---|---|---|---|---|
| | n=1000 | n=5000 | n=7000 | Average | n=1000 | n=5000 | n=7000 | Average |
| SFT | 32.6 | 28.2 | 30.8 | 30.5 | 71.3 | 71.6 | 71.0 | 71.3 |
| Buffer-and-Reinforce (Ours) | **8.9** | **8.6** | **9.3** | **8.9** | **76.4** | **76.6** | **77.1** | **76.7** |

QR-based merging is designed to remove ReinforceLoRA components that interfere with the UserLoRA subspace, leading to a better safety-utility trade-off.

## C. Discussion

### C.1. SafeLoRA vs. QR-Based Merging

SafeLoRA and our QR-based merging strategy target different merging objectives. SafeLoRA derives its safe subspace from the weight difference between the Instruct and Base models, so the resulting subspace can retain substantial utility information in addition to safety-related components. Projecting UserLoRA onto this subspace can therefore preserve utility while suppressing unsafe behavior.

In contrast, ReinforceLoRA in our framework is specialized for safety reinforcement, and our QR-based merging incorporates it in a utility-preserving manner. Since BufferLoRA-based fine-tuning already prevents most harmful updates from being encoded into UserLoRA, the key objective at the merging stage is to inject safety from ReinforceLoRA while minimally interfering with UserLoRA's utility.

To quantify this geometric difference, we measure layer-averaged Energy Retain and Energy Damage. For each layer $\ell$, let $W_U^\ell$ denote the original UserLoRA update and let $\hat{W}^\ell$ denote the projected or merged update. We first compute the projection of $\hat{W}^\ell$ onto the original UserLoRA direction:

$$\text{Proj}_{W_U^\ell}(\hat{W}^\ell) = \frac{\left\langle \text{vec}(\hat{W}^\ell), \text{vec}(W_U^\ell) \right\rangle}{\left\| \text{vec}(W_U^\ell) \right\|_2^2} W_U^\ell. \quad (17)$$

Energy Retain measures how much energy is preserved along the original UserLoRA direction:

$$\text{Energy Retain}^\ell = \frac{\left\| \text{Proj}_{W_U^\ell}(\hat{W}^\ell) \right\|_F^2}{\left\| W_U^\ell \right\|_F^2}. \quad (18)$$

Energy Damage measures the corresponding loss of

*Table A11.* **Ablation study under benign-only user fine-tuning.** We evaluate the contribution of each component when the user data contains no harmful samples ($p = 0$).

| Row | BufferLoRA | ReinforceLoRA | QR | HS ↓ | FA ↑ |
|---|---|---|---|---|---|
| 1 | X | X | X | 32.6 | 71.3 |
| 2 | O | X | X | 17.8 | 75.1 |
| 3 | O | O | X | 4.3 | 74.1 |
| 4 | X | O | X | 3.8 | 68.9 |
| 5 | X | O | O | 8.5 | 75.5 |
| 6 | O | O | O | **8.9** | **76.4** |

UserLoRA-aligned energy:

$$\text{Energy Damage}^\ell = \max\left(0,\ 1 - \text{Energy Retain}^\ell\right). \quad (19)$$

We report the averages of $\text{Energy Retain}^\ell$ and $\text{Energy Damage}^\ell$ across layers in Table A9.

Table A9 shows that directly projecting UserLoRA onto the ReinforceLoRA subspace removes most energy aligned with the original UserLoRA direction, leading to severe utility degradation. By contrast, QR-based merging achieves Energy Retain greater than 1 because its soft orthogonalization preserves the UserLoRA direction while retaining ReinforceLoRA components aligned with it.

Thus, projection-based merging is suitable when the target subspace contains both safety and utility information, whereas our setting requires preserving the user-task subspace while adding a safety-specialized adapter. This explains why QR-based merging better matches the asymmetric objective of our Buffer-and-Reinforce framework.

### C.2. Analysis of Fine-tuning Accuracy Improvements

To better understand why our method improves FA even when $p = 0$, we first examine whether the gain can be explained by insufficient user data in the SFT baseline, by evaluating larger user-data settings with 5,000 and 7,000 GSM8K samples.

As shown in Table A10, increasing the user data size does

not close the FA gap between SFT and our method. With 5,000 and 7,000 user samples, SFT achieves 71.6 and 71.0 FA, whereas our method achieves 76.6 and 77.1 FA, respectively. Moreover, our method consistently achieves substantially lower HS across all data sizes. These results suggest that the improvement is unlikely to be explained solely by insufficient user data in the SFT baseline.

We then conduct an additional ablation study under benign-only user fine-tuning to identify which components contribute to the FA gain. Table A11 shows that Buffer-LoRA alone improves FA from 71.3 to 75.1, whereas Rein-forceLoRA alone does not improve FA, achieving 68.9. This suggests that BufferLoRA itself contributes to the utility gain during user fine-tuning. A plausible explanation is that benign-only fine-tuning can still induce safety-degrading updates, while BufferLoRA stabilizes fine-tuning by suppressing such directions without disrupting utility-relevant gradients. As a result, FA can improve even when $p = 0$.

In addition, ReinforceLoRA with QR-based merging further improves FA to 76.4. Since ReinforceLoRA is trained on both harmful-refusal pairs and benign query-answer pairs, it can contain utility-relevant components in addition to safety-related ones. Our QR-based merging preserves these useful components through soft orthogonalization while injecting safety, which can maintain or even improve downstream utility. This is also consistent with the energy-based analysis in Table A9. Thus, the improvement reflects the intended post-fine-tuning effect of our framework.

## D. LLM Usage

Large Language Models (ChatGPT-5.2) were used solely to improve grammar, clarity, and writing quality. They did not contribute to research ideation, methodological design, experimental execution, data analysis, or interpretation of results.

