# OpenReview forum: "Jailbreak to Protect: Buffering and Reinforcing via Temporary Jailbreaking for Safe Fine-Tuning in Large Language Models"
_ICML.cc/2026/Conference — ICML 2026 spotlight_

### Official Review · Reviewer_pEvj · 2026-02-21

**Soundness:** 4
**Presentation:** 4
**Significance:** 2
**Originality:** 3
**Overall Recommendation:** 4
**Confidence:** 4

**Summary:**

This paper proposes a framework to mitigate against model vulnerabilities to fine-tuning to degrade safety protections. The authors introduce a technique to neutralise harmful updates via temporary harmful gradient saturation during the fine-tuning process (BufferLoRA). This module is used to temporarily jailbreak the model during user fine-tuning, after which a pre-trained "SafetyLoRA" is merged with the "UserLoRA" via a QR decomposition-based strategy to restore and improve safety. The results show that this method leads to SOTA performance across a range of models (Gemma, Qwen, Llama), and does not require additional safety SFT data.

**Compliance With Llm Reviewing Policy:**

Affirmed.

**Final Justification:**

Per rebuttal discussion with authors, I have maintained my score of 4. I think the paper has many merits, but have not elected for a stronger accept because of the value of this work primarily being for FaaS, which is conducted with large models out of scope of this paper.

It is understandable that this work cannot provide evidence of the impact on these models due to computational constraints and the need for white-box access to models. However, this limits the potential impact and applicability of the work.

**Key Questions For Authors:**

1. Is there any evidence to suggest that this method remains effective against models which show more general robustness to jailbreaks?
2. For larger or more robust models, would the data required for BufferLoRA to be effective grow prohibitively? (therefore reducing the impact of the claim that less data is required vs. other techniques?)

**Limitations:**

The authors do not appear to have included an impact statement within the paper. As this paper relies on successfully jailbreaking models, discussion of the impact of exploiting this vulnerability would be valuable.

**Strengths And Weaknesses:**

**Strengths**
1. **Novelty.** The paper provides a novel solution to the problem of adversarial fine-tuning.
2. **Data requirements.** Unlike other solutions to this problem which often involve the need for high quality safety data to include in fine-tuning processes, this work does not require any additional safety data.
3. **Generalisation.** The authors show the effectiveness of the method across multiple model families and using multiple tasks. Demonstrating that results are also effective using datasets other than LAT, and whilst the impact is reduced, the results are still effective versus other techniques.
4. **Results.** In general, the results are very strong compared to other approaches to mitigating the adversarial fine-tuning vulnerability within models.

**Weaknesses**
1. **Reliance on effective jailbreaking.** The work is heavily reliant on the success of the BufferLoRA process. The authors successfully show this effectiveness across models but are reliant on the ability for these models to be jailbroken in the first place. For this work to have long-term impact models must continue to be able to be jailbroken with relative ease. Whilst this may be true for smaller models, this may not be true if models are more robust to jailbreaking attacks.
2. **Scalability.** As the paper highlights, fine-tuning as a service is frequently deployed for frontier model providers (Google, OpenAI), with large models deployed. Whilst the paper shows that this method is effective for the 4B-9B parameter model range, if it is aiming to assist FaaS providers, demonstrating effectiveness on models of a greater scale would be helpful. However, this may not be possible due to computational costs and model access.

---

> ### Author Rebuttal · Authors · 2026-03-31
>
> We thank the reviewer for the thoughtful and encouraging feedback. We especially appreciate the recognition of the novelty, strong empirical results, and broad evaluation across model families and tasks. We also value the concerns regarding reliance on effective jailbreaking and scalability to larger frontier models, which we address below and will clarify in the camera-ready version.
>
> ### **Effectiveness on More Safety-Aligned and Larger Models (W1, W2, Q1, Q2)**
>
> As discussed in our Limitations section, we agree that our method relies on the ability to jailbreak the model. However, the jailbreak setting considered in our work is substantially easier than standard inference-time jailbreak attacks, because BufferLoRA is trained through harmful fine-tuning in a white-box setting, where the model weights can be directly updated.
>
> Moreover, prior work has shown that safety alignment can often be weakened with relatively small amounts of data and training. Consistent with this observation, our experiments also show stable performance under a limited data budget, suggesting that the method does not require excessive harmful or safety data within the evaluated scale.
>
> **Table R8. Results across models with strengthened safety alignment and a larger 13B model under the same training budget (3 epochs, 5K harmful data, and 5K safety data).**
>
> |Method|BufferLoRA HS|SafetyLoRA HS|Final HS|Final FA|
> |:-|:-:|:-:|:-:|:-:|
> |Llama3-8B-Instruct Base|-|-|19.2|62.8|
> |Llama3-8B-Instruct SFT|-|-|75.2|68.4|
> |Llama3-8B-Instruct Ours|88.2|4.3|8.4|77.5|
> |Llama3-8B-LAT Base|-|-|4.1|73.5|
> |Llama3-8B-LAT SFT|-|-|74.0|70.6|
> |Llama3-8B-LAT Ours|87.6|2.1|3.1|75.2|
> |Llama3-8B-ReFAT Base|-|-|16.2|40.2|
> |Llama3-8B-ReFAT SFT|-|-|74.0|68.9|
> |Llama3-8B-ReFAT Ours|87.3|2.2|3.5|71.1|
> |Llama2-13B-Chat Base|-|-|6.4|28.8|
> |Llama2-13B-Chat SFT|-|-|33.2|33.1|
> |Llama2-13B-Chat Ours|85.7|1.5|4.3|35.4|
>
> As shown in Table R8, under the same training budget of 3 epochs, 5,000 harmful samples, and 5,000 safe samples, our framework remains effective across multiple models whose safety has been further strengthened by prior safety-reinforcing methods (LAT [1], ReFAT [2]), as well as a larger 13B model (Llama2-13B-Chat). While this does not fully establish effectiveness for arbitrarily robust frontier models, it does provide evidence that our framework remains effective even when the base model safety has already been substantially strengthened. Importantly, within the 4B-13B range we evaluate, we do not observe a need to increase the harmful/safe data budget as model size or safety alignment strength increases. However, due to resource constraints, we were not able to conduct experiments on models larger than 13B in this work.
>
> We do not claim that robustness or scale is irrelevant in general. Larger or more robust frontier models may require more harmful finetuning steps or data, and scaling to such models remains future work. We will clarify this scope and include these results in the camera-ready version.
>
> [1] Sheshadri, Abhay, et al. "Latent Adversarial Training Improves Robustness to Persistent Harmful Behaviors in LLMs." Transactions on Machine Learning Research.
>
> [2] Yu, Lei, et al. "Robust LLM safeguarding via refusal feature adversarial training." The Thirteenth International Conference on Learning Representations (2025).

---

> > ### Author Rebuttal · Reviewer_pEvj · 2026-04-01
> >
> > I appreciate the authors providing further evidence of the method's effectiveness on safety-aligned Llama variants of various sizes.
> >
> > However, I have not adjusted my score because my concern is that this work aims to be applicable to Fine-tuning-as-a-Service offerings. The paper positions the impact of the work around FaaS and acknowledges that FaaS is popularised by large labs deploying large models, yet this work does not provide evidence of this work being applicable to larger, more robust models.
> >
> > It is understandable that this work cannot provide evidence of the impact on these models due to computational constraints and the need for white-box access to models. However, this limits the potential impact and applicability of the work, as the jailbreaking approach may not work for frontier models where this could have most impact, so I have not changed my recommendation (4).
> >
> > Note: the lack of impact statement remains an issue that should be resolved before publication.
> >
> > Thank you for the paper, and for your response.

---

> > > ### Author Response · Authors · 2026-04-07
> > >
> > > We thank the reviewer for the careful follow-up and for recognizing our additional evidence on safety-aligned Llama variants of different sizes.
> > >
> > > We also appreciate the reviewer's clarification regarding the remaining concern. We understand the reviewer's point that, although our paper is motivated by the Fine-tuning-as-a-Service (FaaS) setting, we do not provide direct evidence on larger frontier models, where the practical impact of such risks could be more significant. We agree that this is a limitation of the current work.
> > >
> > > As the reviewer noted, evaluating such settings is challenging due to computational cost and the need for white-box access to proprietary large models. In the revised manuscript, we will make this limitation more explicit and further moderate our claims about applicability to FaaS scenarios accordingly. We will also add a clear impact statement to discuss both the practical relevance of the threat and the current boundary of our empirical evidence.
> > >
> > > We sincerely appreciate the reviewer's thoughtful feedback and will make sure these concerns are clearly addressed in the camera-ready version.

---

### Official Review · Reviewer_5MBQ · 2026-02-24

**Soundness:** 3
**Presentation:** 3
**Significance:** 2
**Originality:** 4
**Overall Recommendation:** 5
**Confidence:** 4

**Summary:**

This paper presents BufferLoRA, a strategy for Fine-tuning-as-a-Service (FaaS) providers to mitigate users from degrading the safety alignment of LLMs. The procedure involves the FaaS provider learning two private LoRAs for their model, one trained on harmful content to "jailbreak" the LLM (BufferLoRA), and another trained to correct the safety alignment of the jailbroken version (SafetyLoRA). During fine-tuning of a user's LoRA, the provider applies the BufferLoRA to the model so that gradients corresponding to harmful content are mostly saturated, as the BufferLoRA already contains those capabilities. During the process, the user LoRA should learn the residual "safe" behaviors. After fine-tuning, the components of the SafetyLoRA orthogonal to the user LoRA are merged with the user LoRA to mitigate safety reduction caused by general fine-tuning. Experiments and ablations demonstrate the effectiveness of this approach.

**Compliance With Llm Reviewing Policy:**

Affirmed.

**Final Justification:**

I recommend accepting this paper. The discussion period reinforced my positive score, with the authors fully addressing my limited concerns.

**Key Questions For Authors:**

1) Is Figure 1 a hypothetical or was it actually derived by real loss landscapes projected to 2-dimensions? If it is hypothetical, then the language suggesting that insights were derived from the figure should be adjusted (lines 78 and in section 4).

2) Why does BufferLoRA result in a higher FA over SFT when the poison ratio is 0? This may suggest that the user datasets are too small to saturate the capacity of the userLoRA. Table 2 only goes out to 2500 samples, whereas SafetyLoRA sees 5000 benign samples.

**Limitations:**

yes

**Strengths And Weaknesses:**

**Strengths**

Soundness:

1) The paper's claims are well supported, with the approach demonstrating significant mitigation of safety degradation while maintaining (or improving) the fine-tuning accuracy on the benign task.

Presentation:

2) The paper is well written and easy to follow.
3) The tables and figures are informative and visually appealing.

Signifiance:

4) Preventing harm is an important research area, and the BufferLoRA idea is immediately practical to FaaS providers. The proposed approach for privately "jail-breaking" a model in conjunction with fine-tuning is likely to result in many new research directions.

Originality:

5) The main idea is clever and novel. It also provides an orthogonal defense to other approaches, offering a suite of options for FaaS providers.

**Weaknesses**

Soundness:

1) The fact that BufferLoRA outperforms SFT with p=0 may suggest the user datasets are too small to saturate the user LoRA, and that your method is getting an unfair boost from the data used to train the SafetyLoRA. The ablation study is good, but I'd like to see it repeated with p=0 to better understand this phenomenon, especially since most FaaS users will not be trying to reduce safety.

Presentation:

2) It was a bit unclear whether Figure 1 is real or just a motivating example. If it is just a motivating example, then it should be stated clearly.

Signifiance:

3) The approach is only applicable to FaaS providers seeking to mitigate the safety degradation of their LLMs. While important, the scope is limited.

Originality:

N/A

---

> ### Author Rebuttal · Authors · 2026-03-31
>
> We thank the reviewer for the careful and constructive feedback. We address the concerns below and will incorporate the clarifications discussed in this rebuttal into the camera-ready version.
>
> ### **Analysis on the p = 0 Setting and User Data Size (W1, Q2)**
>
> To better understand why our method improves FA even when p = 0, we additionally conducted a p = 0 ablation study (Table R5).
>
> **Table R6. Ablation study under benign-only user fine-tuning (p=0).**
>
> | BufferLoRA | SafetyLoRA | QR |  HS  |  FA  |
> | :--------: | :--------: | :-: | :--: | :--: |
> |     X     |     X     | X | 32.6 | 71.3 |
> |     O     |     X     | X | 17.8 | 75.1 |
> |     O     |     O     | X | 4.3 | 74.1 |
> |     X     |     O     | X | 3.8 | 68.9 |
> |     X     |     O     | O | 8.5 | 75.5 |
> |     O     |     O     | O | 8.9 | 76.4 |
>
> Table R6 shows that the FA gain cannot be explained solely by the additional benign data used to train SafetyLoRA. BufferLoRA alone already improves FA from 71.3 to 75.1, while SafetyLoRA alone does not improve FA (68.9). This suggests that one source of the gain is the effect of BufferLoRA during user fine-tuning. A plausible interpretation is that, since benign-only fine-tuning can still induce safety-degrading updates, BufferLoRA stabilizes user fine-tuning by regularizing safety-degrading directions while preserving utility directions, which can improve FA even when p = 0.
>
> We can also observe that SafetyLoRA + QR-based merging further improves the final performance (FA 76.4). Since SafetyLoRA is trained on both harmful-refusal pairs and benign query-answer pairs, it can contain utility-relevant components in addition to safety-related ones. The soft orthogonalization in our QR-based merging preserves such utility-relevant components, helping maintain or even improve downstream utility. We therefore view this improvement as part of our intended post-fine-tuning contribution, rather than as an unfair advantage from auxiliary data alone.
>
> To further address the reviewer's concern about user data size, we also evaluated 5K and 7K user data settings.
>
> **Table R7. Results on larger user-data settings (5K and 7K).**
>
> |Method|GSM8K (n=1K) HS|GSM8K (n=1K) FA|GSM8K (n=5K) HS|GSM8K (n=5K) FA|GSM8K (n=7K) HS|GSM8K (n=7K) FA|
> |:-|:-:|:-:|:-:|:-:|:-:|:-:|
> |SFT|32.6|71.3|28.2|71.6|30.8|71.0|
> |Ours|8.9|76.4|8.6|76.6|9.3|77.1|
>
> As shown in Table R7, increasing the user data size does not materially close the gap between SFT and our method: SFT achieves FA 71.6 / 71.0, whereas our method achieves FA 76.6 / 77.1 for 5K / 7K, respectively. This suggests that the improvement is unlikely to be explained solely by insufficient user data size.
>
> In the camera-ready version, we will clarify this point explicitly by explaining that the FA gain at p = 0 is not solely attributable to extra benign data for SafetyLoRA or to insufficient user data size, but is also consistent with the intended roles of BufferLoRA and our post-fine-tuning merging design.
>
> ---
> ### **Clarification of Figure 1 (W2, Q1)**
>
> We thank the reviewer for pointing out this ambiguity. Figure 1 is not a hypothetical illustration, but is derived from a real loss-landscape analysis. Specifically, we construct it using Llama3-8B-Instruct as the safety-aligned LLM, and a jailbroken model obtained by LoRA fine-tuning Llama3-8B-Instruct on 5,000 harmful samples from LAT. For evaluation, we use 100 harmful examples from BeaverTails and 100 harmless examples from GSM8K. We visualize the loss landscape using the `loss_landscapes` package by projecting the loss surface onto a 2D random plane around the current model parameters.
>
> In the camera-ready version, we will clarify in both the caption and the main text that Figure 1 is an empirical 2D projected loss landscape rather than a schematic illustration.
>
> ### **Scope and Applicability of the Proposed Framework (W3)**
>
> We agree that our framework is designed for a specific setting, namely fine-tuning safety-aligned LLMs in the FaaS scenario. This is a deliberate problem formulation rather than a claim of universal applicability.
>
> At the same time, we believe this setting is practically important. Safety degradation after fine-tuning a safety-aligned LLM is a meaningful and well-recognized problem, and many prior works have been proposed to address it. Our work is positioned within this line of research, with a particular focus on the FaaS setting where a provider must preserve safety while enabling user-specific customization.
>
> While the overall framework is optimized for FaaS, some components may also have broader relevance beyond this setting. For example, our QR-based merging can be viewed more generally as a LoRA merging strategy that prioritizes preserving one adapter while reducing interference from another. We will clarify this scope and positioning more explicitly in the camera-ready version.

---

> > ### Author Rebuttal · Reviewer_5MBQ · 2026-03-31
> >
> > The authors have adequately addressed my concerns, confirming that Fig 1 is derived from a real example and provided the requested ablation against SFT with p=0. I appreciate their analysis and interpretation of the new results. The authors also offer a potential avenue for increasing the scope of their findings.

---

> > > ### Author Response · Authors · 2026-04-07
> > >
> > > We thank the reviewer for the encouraging feedback and for confirming that our rebuttal has adequately addressed the concerns. We are pleased that the clarification that Figure 1 is based on a real example and the additional ablation against SFT with p=0 were helpful.
> > >
> > > We also appreciate the reviewer's positive comments on our analysis and interpretation of the new results. In camera-ready version, we will incorporate these clarifications more clearly, and further discuss the broader implications and possible extensions suggested by this rebuttal.
> > >
> > > We are grateful for the reviewer's thoughtful feedback, which has helped strengthen the paper.

---

### Official Review · Reviewer_dUXo · 2026-03-02

**Soundness:** 3
**Presentation:** 3
**Significance:** 2
**Originality:** 2
**Overall Recommendation:** 4
**Confidence:** 5

**Summary:**

This paper proposes a post-fine-tuning stage defense for harmful fine-tuning utilzing some ideas on model merging.

**Compliance With Llm Reviewing Policy:**

Affirmed.

**Final Justification:**

Concerns are fully adressed but the authors should definitely discuss [1] that I mention in camera ready. Please consider what the authors of [1] will feel if they see this paper without properly citing and discussing them.

**Key Questions For Authors:**

The model merging method proposed in post-fine-tuning stage is interesting and would be meaningful if a good comparison  and discussion with SafeLoRA is conducted. However, I don't understand why the authors ignore the existing contribution from Security Vector[1]. You can say you are further optimizing the post-fine-tuning merging part of Security Vector, but why instead choosing to not discuss or not knowing it?

Very likely I will not increase my score  after rebuttal (or even lower the score) given that the contribution of this paper is unclear, no matter from the perspective of advancing security Vector[1] or advancing Safe Lora[3].  While I give 4(weak accept), the real score I give should be 3.5  (no lean to accept or reject).

[1] Making Harmful Behaviors Unlearnable for Large Language Models

[3] Safe LoRA: the Silver Lining of Reducing Safety Risks when Fine-tuning Large Language Models

**Limitations:**

Yes

**Strengths And Weaknesses:**

Strength:

1. Extensive literature review is conducted though one highly relevant research is not discussed.
2. Comprehensive experiment is conducted.
3. The method makes perfect senses.

Weakness:
1. Two highly relevant research  Security Vector [1] and Panacea [2] are not discussed and a line of research with highly similar inisght is not properly discuss. The authors should add discussion on this on the methodology part.

* The first method to exploit the idea of temporaily jailbreaking model with LoRA is exploited by Security Vector [1], published in 2023. This method is highly similar and actually are almost identical to the idea of the propose method Buffer LORA. Specifically, Eq. (2) in BufferLoRA is 100% identical Eq. (3) in [1], which is conducted before fine-tuning. Eq. (4) in Buffer LORA is 100\% identical Eq. (4) in [1] conducted during fine-tuning.  After fine-tuning, the two papers are slightly different. If you disagree with this point, feel free to challenge me in the rebuttal with evidence and I am happy to discuss. If you agree their equivalence, could you explain why [1] is not cited in the initial submission?  And please add discussion in the methodology section to mention this similarity with the prior work.

[1] Making Harmful Behaviors Unlearnable for Large Language Models

*  A similar idea is studied by Panacea[2]. Panacea also make the fine-tuning weights to be harmful (i..e., low harmful loss) before post-fine-tuning perturbing. But it does not optimize the harmful LoRA before fine-tuning. Instead, during fine-tuning, it jointly train the recover perturbation along with the user LORA, such that the recover perturbation after fine-tuning can maximally increase the harmful loss for the user LORA.

 [2] Panacea: Mitigating harmful fine-tuning for large language models via post-fine-tuning perturbation


2. Excluding the contribution of Eq. (2) and Eq. (4), the real contribution should be Section 4. 3, which merges  UserLoRA  and SafetyLoRA  via QR decomposition. This contribution seems to be novel. However, there is no comparison results that shows that this contribution empirically advance existing method security vector[1], which only remove the BufferLoRA from the model.

3. In section 4.3, the authors merge the safety update with user update by projecting the safety weights to an orthogonal subspace of the user weights. However, this method share some similarity with SafeLoRA and is not properly discussed as well. This operation indeed seems to be contradictory to SafeLORA[3]. Specifically, Safe LORA project the user update into the safety subspace but this paper project the safety update into the **orthgonal of user subspace**. Could you explain why this contradition happens and which should be a correct projection way with emprical data?  Specifically, whether and why we should project to the orthogonal subsapce or not. Please note simple comparision (e.g., harmful score and finetune accuracy) of two methods  is fine, but is not very convincing to me.


[3] Safe LoRA: the Silver Lining of Reducing Safety Risks when Fine-tuning Large Language Models

---

> ### Author Rebuttal · Authors · 2026-03-31
>
> We appreciate the reviewer's valuable comments. Below we address the main concerns and will incorporate these clarifications into the camera-ready version.
> ### **Discussion and Comparison with Security Vector and Panacea (W1, W2, Q1)**
>
> We sincerely thank the reviewer for pointing out the relevance of Security Vector and Panacea. We agree that our method has a high-level overlap with Security Vector in the temporary harmful adapter stage. We acknowledge that we missed this paper during our prior work survey, and this omission was not intentional. In the camera-ready version, we will explicitly discuss both methods in the method and related work sections, and refine the positioning of our contribution accordingly.
>
> That said, our paper still makes distinct contributions in interpretation, analysis and post-finetuning design. While Security Vector is primarily described from the perspective of consistency with harmful target responses, we reinterpret the temporary harmful adapter through the lens of gradient saturation. Specifically, we empirically show that it reshapes the optimization landscape so that harmful gradients become saturated while downstream-task gradients are preserved. Such a gradient-based analysis is not provided in Security Vector.
>
> We also agree that the clearest technical novelty of our paper lies in the post-finetuning stage. Security Vector mainly focuses on preserving the original safety level during finetuning, whereas we introduce SafetyLoRA and QR-based merging to further reinforce safety after finetuning while minimizing interference with downstream utility. Moreover, SafetyLoRA is trained in an adversarial-training-like setup under a temporary jailbroken state, encouraging stronger refusal behavior by overcoming the temporary jailbreaking itself.
>
> Beyond our ablation study in Table 6, we provide a direct comparison with Security Vector.
>
> **Table R4. Comparison with Security Vector.** "w/o KL" denote variants without the KL regularization in Security Vector, which is used to reduce the effect of the Security Vector on non-target behaviors.
> |Method|HS|FA|
> |:-|:-:|:-:|
> |SFT|75.2|68.4|
> |Security Vector|20.5|73.4|
> |Security Vector (w/o KL)|18.0|75.4|
> |SafetyLoRA+QR|21.8|56.6|
> |Ours|8.4|77.5|
>
> Table R4 shows that our gain comes not only from the temporary harmful adapter but also from the post-finetuning design, which further improves the safety-utility trade-off beyond Security Vector. Importantly, our design is built upon the temporary harmful adapter: QR-based merging works best when UserLoRA contains mostly utility-related information and little harmful information. Otherwise, harmful and utility-related information become entangled in UserLoRA, making it a poor reference for merging.
>
> We also agree that Panacea should be discussed more explicitly. At a high level, both methods combine a finetuning-time intervention with a post-finetuning mechanism to restore safety. However, their mechanisms differ: Panacea is a perturbation-based restoration method, whereas ours is based on temporary jailbreaking and orthogonal safety merging.
>
> ---
> ### **Differences Between SafeLoRA and QR-Based Merging (W3)**
>
> We would like to clarify that SafeLoRA and our QR-based merging are not contradictory, but are designed for different objectives. SafeLoRA derives its safe subspace from the weight difference between the Instruct and Base models, so this subspace retains substantial utility information in addition to safety. Thus, projecting UserLoRA into this subspace preserves utility while suppressing unsafety. In contrast, our SafetyLoRA is highly specialized for safety. In our framework, BufferLoRA-based finetuning already prevents most harmful updates from being encoded into UserLoRA, so the key objective at merging is to preserve UserLoRA’s downstream-task utility while injecting safety from SafetyLoRA.
>
> To illustrate this geometrical difference, we analyze layer-averaged Energy Retain/Damage, where Energy Retain measures how much energy is preserved along the original UserLoRA direction, while Energy Damage measures the corresponding loss.  Full definitions will be provided in the discussion thread as an Appendix.
>
> **Table R5. Layer-averaged Energy Retain/Damage analysis with HS and FA.**
> |Method|Energy Retain|Energy Damage|HS|FA|
> |:-|:-:|:-:|:-:|:-:|
> |Project to SafetyLoRA|1.20E-05|0.999|18.0|63.3|
> |QR-based (Ours)|1.09|1.31E-03|8.4|77.5|
>
> Table R5 shows that "Project to SafetyLoRA" removes almost all energy aligned with the original UserLoRA direction, explaining its severe utility degradation. In contrast, QR-based merging achieves Energy Retain greater than 1 because, through soft orthogonalization, it not only preserves the original UserLoRA direction but also retains SafetyLoRA components aligned with it. Therefore, the appropriate choice depends on the objective of the merge. We will clarify this distinction in the camera-ready version.

---

> > ### Author Rebuttal · Reviewer_dUXo · 2026-04-03
> >
> > I prefer not to stop this paper from publication given that other reviewers seem to like this work, and there seem to be some contribution on the model merging part after post-fine-tuning. I will keep my score as 4 weak accept.
> >
> > However, I suggest the authors to properly discuss with [1] in the camera ready and the I also want to mention here  for the AC to consider the similarity between [1] and the BufferLoRA method. [1] is not cited in the original paper and this is a bad/unethical practice (if this is intentional).
> >
> > [1] Making Harmful Behaviors Unlearnable for Large Language Models

---

> > > ### Author Response · Authors · 2026-04-07
> > >
> > > We sincerely thank the reviewer for the careful assessment and for maintaining a score of 4. We appreciate the reviewer's recognition that our work offers a meaningful contribution, particularly in the post-fine-tuning LoRA merging component.
> > > We also sincerely apologize again for missing the important prior work [1]. This omission was unintentional, and we regret that we did not properly cite and discuss it in the original submission.
> > > In the camera-ready version, we will explicitly cite and discuss [1], and carefully clarify both the similarities and the differences between [1] and our framework so that our contribution is positioned more accurately and transparently.
> > >
> > > [1] Zhou, Xin, et al. "Making harmful behaviors unlearnable for large language models." Findings of the Association for Computational Linguistics: ACL 2024. 2024.
> > >
> > > ---
> > > ### **Appendix. Definition of Energy Retain/Damage**
> > >
> > > For each layer $\ell$, let $W_u^{(\ell)}$ denote the original UserLoRA update and $\hat{W}^{(\ell)}$ the projected or merged update. We define
> > >
> > > $\mathrm{Proj}_{W_u^{(\ell)}}(\hat{W}^{(\ell)})=\frac{\langle \mathrm{vec}(\hat{W}^{(\ell)}), \mathrm{vec}(W_u^{(\ell)})\rangle}{\|\mathrm{vec}(W_u^{(\ell)})\|_2^2} \, W_u^{(\ell)},$
> > >
> > > $\text{Energy Retain}^{(\ell)}=\frac{\|\mathrm{Proj}_{W_u^{(\ell)}}(\hat{W}^{(\ell)})\|_2^2}{\|W_u^{(\ell)}\|_2^2},$
> > >
> > > $\text{Energy Damage}^{(\ell)}=\max \left(0,\ 1-\text{Energy Retain}^{(\ell)}\right),$
> > >
> > > We report the averages of $\text{Energy Retain}^{(\ell)}$ and $\text{Energy Damage}^{(\ell)}$ across layers.

---

### Official Review · Reviewer_T6Ch · 2026-03-09

**Soundness:** 2
**Presentation:** 3
**Significance:** 3
**Originality:** 2
**Overall Recommendation:** 4
**Confidence:** 3

**Summary:**

This paper introduces a novel defense paradigm addressing the safety alignment degradation caused by user-provided data in Fine-tuning-as-a-Service (FaaS) scenarios. This study introduces BufferLoRA to intentionally and temporarily jailbreak the model during the user fine-tuning phase. This cleverly leverages the natural saturation of harmful gradients to neutralize safety degradation. Subsequently, the paper utilizes an orthogonal merging strategy based on QR decomposition to integrate the UserLoRA, which captures downstream task knowledge, with a pre-trained SafetyLoRA without interference. The authors provide extensive experiments to demonstrate the defense's effectiveness in keeping the utility and safety alignment.

**Compliance With Llm Reviewing Policy:**

Affirmed.

**Final Justification:**

The authors provide extra experiments and discussions for W1, 2, 3, and 5, which addressed my main concerns.

**Key Questions For Authors:**

Please see the above weaknesses.

**Limitations:**

yes

**Strengths And Weaknesses:**

Strengths:
1. The key observation of saturation of harmful gradients under jailbroken states is  inspiring; the authors also provide nice plots for illustration.

2. The math formulas are rigorously and clearly defined, providing solid theoretical support for orthogonal projection and effective subspace extraction in the interference-free merging strategy.

3. The authors provide extensive experiments, validating the framework's exceptional robustness under extreme data contamination across various base model architectures and complex downstream tasks.

Weaknesses:
Although jailbreak to protect is interesting, the paper lacks certain details, making the approach read more like an empirical method tailored to specific settings rather than a universally applicable solution. To better demonstrate the method's generalizability, the authors should conduct the following experiments and provide further in-depth discussion:

1. the paper arbitrarily applies a fixed threshold (layers 0-15) for safety relying on existing literature. The process for determining the optimal threshold across different LLMs remains unclear.

2. Training BufferLoRA on the LAT dataset is only effective for harmful query-response pairs. Whether this applies to more advanced jailbreak methods, such as trigger-based jailbreaking, remains unknown.

3. How to configure the optimal hyperparameters are missing. Several hyperparameters could significantly impact performance. For instance, how should the training epochs for BufferLoRA, UserLoRA, and SafetyLoRA be configured? Would overfitting in any single module affect the overall defense performance? Additionally, how is the optimal dataset size for each module determined?

Other weaknesses:

4. The authors do not compare the proposed QR decomposition merging with popular PEFT merging methods. Examples are Task Arithmetic [1], TIES-Merging [2], LoRA-LEGO [3] and so on.

5. It is unrealistic to assume that service providers have prior knowledge of the specific attack types. Consequently, providers would need to train multiple LoRA pairs for different jailbreak attacks. Combined with the necessary hyperparameter grid search, the overall computational cost could become significantly high.

> ***References***
>
> [1] Ilharco et al (2023). Editing Models with Task Arithmetic. ICLR 2023.
>
> [2] Yadav et al (2023). TIES-Merging: Resolving Interference When Merging Models. NeurIPS 2023.
>
> [3] Zhao et al (2025). Merging LoRAs like Playing LEGO: Pushing the Modularity of LoRA to Extremes Through Rank-Wise Clustering. ICLR 2025.

---

> ### Author Rebuttal · Authors · 2026-03-31
>
> We thank the reviewer for the constructive feedback. Below we clarify the main points and will reflect them in the camera-ready version.
> ### **Layer Selections for Safety and Utility Gradient Analysis (W1)**
> We agree that layer 15 is not a universal boundary across all LLMs. It is used as a coarse boundary only for analyzing Llama3-8B-Instruct, and a different model could require a different boundary. However, the broader trend remains consistent: as discussed in the Observation section, prior work identifies safety-related layers in the early-to-middle layers, whereas utility-related layers emerge strongly in the middle-to-late layers. Consistently, our full layer-wise analysis on Llama3-8B-Instruct shows that safety-gradient contrast becomes much less pronounced after around layer 15, while utility-related gradients exhibit local spikes in earlier layers. Note that across all experiments, LoRA modules are attached to all layers regardless of the model. The full layer-wise analysis will be added to the Appendix.
>
> ---
> ### **Hyperparameter Configuration and Sensitivity Analysis (W3)**
> We agree that hyperparameter selection is practically important, thereby we examined module sensitivity to training epochs and data size. UserLoRA's data size sensitivity is already reported in Table 2 of the main manuscript.
>
> **Table R1. Hyperparameter sensitivity of BufferLoRA, SafetyLoRA, and UserLoRA.**
>
> Default DataSize/Epochs: BufferLoRA = 5K/3, SafetyLoRA = 5K/3, UserLoRA = 1K/3.
>
> |Ablation|DataSize/Epochs|BufferLoRA HS|SafetyLoRA HS|Final HS|Final FA|
> |:-|:-:|:-:|:-:|:-:|:-:|
> |Default|-|88.2|4.3|8.4|77.5|
> |BufferLoRA Epoch|5K/1|87.9|3.7|10.6|76.6|
> ||5K/10|87.5|6.2|11.7|69.2|
> |SafetyLoRA Epoch|5K/1|88.2|4.9|8.2|74.7|
> ||5K/10|88.2|5.3|9.9|75.2|
> |BufferLoRA DataSize|3K/3|86.8|3.3|7.8|77.1|
> ||10K/3|87.4|3.9|8.2|76.0|
> |SafetyLoRA DataSize|3K/3|88.2|4.9|7.4|74.8|
> ||10K/3|88.2|3.9|8.7|76.6|
> |UserLoRA Epoch|1K/1|88.2|4.3|8.5|76.9|
> ||1K/10|88.2|4.3|9.3|76.7|
>
> Table R1 shows that our framework is fairly robust to moderate hyperparameter variation, although excessive training a single module can degrade the final defense performance. In practice, BufferLoRA and SafetyLoRA should be configured by monitoring HS, whereas UserLoRA is more task-dependent, since its optimal setting depends on the type and amount of user data. We will include these practical guidelines in the camera-ready version.
>
> ---
> ### **Cross-Attack Generalization and Practicality (W2, W5)**
> We would like to clarify that our framework does not assume prior knowledge of a specific jailbreak type. Although BufferLoRA is trained under the harmful finetuning attack, the resulting jailbroken state targets harmful behavior at a broader level rather than overfitting to a particular attack format.
>
> Importantly, in the FaaS setting, harmful finetuning with malicious query-response pairs is itself one of the strongest jailbreak attacks, since the attacker can directly modify model parameters through white-box finetuning by providing harmful user data. To examine robustness beyond this setting, we evaluate our method against trigger-based and advanced jailbreak attacks (GCG, PAIR, TAP).
>
> **Table R2. Cross-attack generalization results.**
> |Method|Backdoor HS|Backdoor FA|GCG HS|PAIR HS|TAP HS|FA|
> |:-|:-:|:-:|:-:|:-:|:-:|:-:|
> |SFT|73.9|68.7|74.7|68.4|53.0|68.4|41.0|68.4|
> |Ours|8.8|76.1|3.1|27.0|20.0|76.1|
>
> Table R2 shows that our method remains robust across diverse jailbreak attacks, suggesting that the method is not merely learning a narrow attack-specific pattern, but instead contributes to broader harmful behavior suppression.
>
> These results also address the practicality concern in W5. A single BufferLoRA/SafetyLoRA pipeline transfers across multiple attack settings, so the framework does not require training separate LoRA pairs per attack. Moreover, our hyperparameter stability in Table R1 alleviates the need for exhaustive grid search in practice.
>
> ---
> ### **Comparison with Representative PEFT Merging Methods (W4)**
> We agree that comparison with representative PEFT merging methods is important. Therefore, we compared our QR-based merging with Task Arithmetic, TIES-Merging, and LoRA-LEGO.
>
> **Table R3. Comparison with representative PEFT merging baselines.**
> |Method|HS|FA|
> |:-|:-:|:-:|
> |TaskArithmetic|12.4|70.9|
> |TIES-Merging|7.9|74.8|
> |LoRA-LEGO|6.4|74.2|
> |Ours|8.4|77.5|
>
> Table R3 shows that our method achieves the best overall safety-utility trade-off, as baselines with slightly lower HS incur a clearer cost in FA. This is consistent with our motivation. In our setting, UserLoRA already contains little harmful information due to BufferLoRA-based finetuning, so the key objective at the merging stage is to preserve the UserLoRA's utility while injecting SafetyLoRA's safety with minimal interference. Generic PEFT merging methods do not explicitly account for this asymmetric priority between the two adapters, whereas our QR-based merging is designed for precisely this purpose.

---

> > ### Author Rebuttal · Reviewer_T6Ch · 2026-04-01
> >
> > Most of my concerns have been adequately addressed. Please ensure these experiments and discussions (especially W1,2,3,5) are added to the revised manuscript. I will raise my score to 4.

---

> > > ### Author Response · Authors · 2026-04-07
> > >
> > > We truly appreciate the reviewer’s thoughtful reconsideration and for increasing the score to 4. We are pleased that our rebuttal was able to resolve most of the concerns.
> > >
> > > We also thank the reviewer for recommending that the additional experiments and discussions be incorporated into the revised manuscript. In the camera-ready version, we will integrate the key clarifications associated with W1, W2, W3, and W5, so that the paper more fully captures the supporting evidence and discussion developed in the rebuttal.
> > >
> > > We are grateful for the constructive feedback, which has helped us improve the presentation and overall completeness of the paper.

---

### Decision · Program_Chairs · 2026-04-30

**Decision:**

Accept (spotlight)

**Comment:**

There is a general consensus among four reviewers on the acceptance of this paper. The proposed buffer LoRA approach is neat and shown to be effective in experiments. Reviewers are engaged in discussions and are satisfied with the authors’ responses. Reviewers are hoping to see more discussion on the real-world application of the proposed method: larger model, more realistic setting etc. while reviewers also acknowledge these are not easily actionable for academic research. I would encourage the authors to take the feedback into consideration to further improve the draft.